# Individually Fair Diversity Maximization

**Ruien Li**
School of Data Science and Engineering
East China Normal University
Shanghai, China
reli@stu.ecnu.edu.cn

**Yanhao Wang**\*
School of Data Science and Engineering
East China Normal University
Shanghai, China
yhwang@dase.ecnu.edu.cn

## Abstract

We consider the problem of diversity maximization from the perspective of individual fairness: given a set $P$ of $n$ points in a metric space, we aim to extract a subset $S$ of size $k$ from $P$ so that (1) the diversity of $S$ is maximized and (2) $S$ is *individually fair* in the sense that every point in $P$ has at least one of its $\frac{n}{k}$-nearest neighbors as its "representative" in $S$. We propose $(O(1), 3)$-bicriteria approximation algorithms for the individually fair variants of the three most common diversity maximization problems, namely, max-min diversification, max-sum diversification, and sum-min diversification. Specifically, the proposed algorithms provide a set of points where every point in the dataset finds a point within a distance at most 3 times its distance to its $\frac{n}{k}$-nearest neighbor while achieving a diversity value at most $O(1)$ times lower than the optimal solution. Numerical experiments on real-world and synthetic datasets demonstrate that the proposed algorithms generate solutions that are individually fairer than those produced by unconstrained algorithms and incur only modest losses in diversity.

## 1 Introduction

As machine learning (ML) algorithms are widely used in automated decision-making processes such as banking, hiring, and education, concerns have been raised about their negative social consequences [30, 32], e.g., discriminatory treatment against specific individuals. Recently, there has been extensive literature on *algorithmic fairness*, aiming to define the notion of fairness in ML problems and design effective and efficient fairness-aware algorithms [10, 31]. Such considerations have been taken into account in many data-driven ML tasks, including classification [12], clustering [44], ranking [34, 48], matching [41], and data summarization [8, 23].

This paper focuses on the diversity maximization problem and addresses its individually fair variants. *Diversity maximization* is a fundamental combinatorial optimization problem with broad applications in feature selection [47], search [15, 42], and recommendation [4, 7]. Generally, its objective is to extract a subset $S$ of size $k$ from a set $P$ of $n$ points so that the diversity of $S$ (measured by the point-wise dissimilarity) is maximized. The existing studies on diversity maximization primarily consider three objectives, namely *max-min diversification*, which aims to maximize the minimum distance between any pair of selected points, *max-sum diversification*, which aims to maximize the sum of pairwise distances between selected points, and *sum-min diversification*, which aims to maximize the sum of the minimum distances from each selected point to its nearest neighbor among the other selected points. All three problems have been extensively investigated in the literature [6, 9, 20, 27], where they are also, respectively, referred to as *remote-edge*, *remote-clique*, and *remote-pseudoforest*. The unconstrained max-min diversification problem is NP-complete in metric spaces, and a greedy algorithm [16] offers a 2-approximation, which has also been shown to be tight [39]. For unconstrained max-sum and sum-min diversification problems, it is impossible to obtain an

---

\*Corresponding Author.

39th Conference on Neural Information Processing Systems (NeurIPS 2025).

approximation factor better than 2 in polynomial time if we assume that the planted clique conjecture holds [6, 22], while existing studies have proposed the best possible 2-approximation algorithm for max-sum diversification through greedy selection [19], and the best-known 8-approximation algorithm for sum-min diversification by randomization and solving linear programs (LPs) [6]. Given that each of these objectives highlights different aspects of diversity, we consider all of them in this paper to obtain a comprehensive understanding of the interplay between diversity and individual fairness in various application scenarios.

Currently, efficient algorithms for diversity maximization under *group fairness* or *partition matroid* constraints have been proposed in [2, 3, 33, 45, 46]. They consider that each point in $P$ denotes an individual associated with a particular demographic attribute, e.g., gender or race. As such, $P$ is divided into different demographic groups, and an algorithm is required to select a subset $S$ from $P$ not only to maximize the diversity of $S$ but also to ensure the selection of a pre-determined number of points from every group in $S$ for equitable representation. Despite these studies, little attention has been paid to diversity maximization under *individual fairness* constraints. Inspired by an individual notion of fairness for the facility location problem [21], the selection of a given point set $P$ is fair if every point in $P$ has a center among its $(|P|/k)$-closest neighbors. Compared with group fairness, individual fairness differs in two crucial aspects. First, it does not require access to group attributes, while group fairness relies on predefined features such as gender, age, or race. Second, the two notions consider different perspectives of fairness. While group fairness emphasizes equitable representation across groups, it may overlook disparities among individuals within each group. Individual fairness, by contrast, operates at the point level to ensure that each individual is adequately represented in the selected subset.

The need for individual fairness, in particular, is natural and reasonable in many scenarios [9, 13, 24, 38, 40]. For instance, consider a fast-food franchise that selects $k$ supply outlets within a city. From the perspective of dispersion and diversity, we need to choose outlet locations that are spread out to avoid over-concentration in one region, thereby increasing overall market coverage and potential revenue. However, such dispersion strategies may still leave many individuals far from any chosen outlet, thus excluding them from convenient services. Incorporating individual fairness can address this issue by ensuring that each resident lies within a reasonable distance of at least one facility so that no individual is neglected. This balance not only enhances potential revenue but also improves the service experience for all customers, underscoring the need to balance diversity objectives with individual fairness considerations.

**Our Contributions.** With these motivations, we study the problem of $\alpha$-fair $k$-selection under individual fairness constraints for the three common diversity maximization problems. Here, the parameter $\alpha \geq 1$ can be viewed as a fairness tolerance level. Larger $\alpha$ values allow a looser notion of fairness and thus potentially more diverse solutions, while $\alpha = 1$ corresponds to the strictest form of individual fairness. Moreover, our algorithms are analyzed under the notion of $(\beta, \gamma)$-*bicriteria approximation*, which means that they may relax the fairness constraints with a factor of $\gamma$ (e.g., allowing each point to be covered within a small multiplicative slack) while still providing provable guarantees on the achieved diversity within a factor of $\beta$. Our main results include the following.

- **Max-Min Diversification:** For any $\alpha \geq 1$ and $k \in \mathbb{Z}_+$, we give a $(\beta, 3)$-bicriteria approximation algorithm for max-min diversification under individual fairness constraints if there exists a $\beta$-approximation algorithm for max-min diversification under partition matroid constraints. Using the approximation algorithm of [46] as a subroutine, the approximation factor can be instantiated as $(5 + \varepsilon, 3)$ for any $\varepsilon > 0$.

- **Max-Sum Diversification:** For any $\alpha \geq 1$ and $k \in \mathbb{Z}_+$, we give a $(\beta(4 + \varepsilon), 3)$-bicriteria approximation algorithm for max-sum diversification under individual fairness constraints if there exists a $\beta$-approximation algorithm for max-sum diversification under partition matroid constraints. Using the 2-approximation local search algorithm in [2] as a subroutine, the approximation factor can be instantiated as $(8 + \varepsilon, 3)$ for any $\varepsilon > 0$.

- **Sum-Min Diversification:** For any $\alpha \geq 1$ and $k < n/3$, we give a $(\beta(4 + \varepsilon), 3)$-bicriteria approximation algorithm for sum-min diversification under individual fairness constraints if there exists a $\beta$-approximation algorithm for sum-min diversification under partition matroid constraints. Using the LP-based 8-approximation algorithm in [6] as a subroutine, the approximation factor can be instantiated as $(32 + \varepsilon, 3)$ for any $\varepsilon > 0$.

To the best of our knowledge, these are the first algorithms with theoretical guarantees for individually fair max-min, max-sum, and sum-min diversification problems. Finally, we evaluate the empirical performance of our algorithms on real-world and synthetic data sets. The results demonstrate that our algorithms generate solutions that are individually fairer than those produced by unconstrained algorithms while incurring modest diversity losses. In addition, our algorithms also exhibit high time efficiency and scalability.

## 2   Related Work

**(Unconstrained) Diversity Maximization.**   Diversity maximization has been studied from a graph-theoretic perspective since the 1990s. Ravi et al. [39] proposed a 2-approximation algorithm for max-min diversification and proved that the bound cannot be tighter unless P = NP. Hassin et al. [19] proposed a 2-approximation algorithm for max-sum diversification, and Bhaskara et al. [6] further proved that the approximation factor cannot be improved under the planted clique assumption [22]. Chandra and Halldórsson [9] proposed a $O(\log n)$-approximation algorithm for sum-min diversification. Bhaskara et al. [6] further proposed an 8-approximation algorithm for sum-min diversification and proved that the factor cannot be better than 2 under the same planted clique assumption. These algorithms cannot be used for diversity maximization variants with group or individual fairness constraints. Nevertheless, they often serve as building blocks for the design of fair diversity maximization algorithms.

**Diversity Maximization under Group Fairness Constraints.**   Moumoulidou et al. [33] first proposed approximation algorithms for the group-fair variant of max-min diversification. Addanki et al. [3] improved the approximation ratios of the algorithms in [33]. Wang et al. [45] proposed streaming algorithms for the group-fair variant of max-min diversification. Wang et al. [46] developed an exact algorithm and a $(5 + \varepsilon)$-approximation algorithm for max-min diversification with bounded-size group fairness constraints. Abbassi et al. [2] proposed 2-approximation local-search algorithms for max-sum diversification under matroid constraints. Bhaskara et al. [6] proposed an 8-approximation algorithm for sum-min diversification under matroid constraints based on randomization and linear programming. Mahabadi and Trajanovski [28] proposed two deterministic algorithms for sum-min diversification under group fairness constraints. One is a $O(m \cdot \log k)$-approximation algorithm with an exponential time complexity w.r.t. $k$, and the other is a $O(m^2 \cdot \log k)$-approximation algorithm that runs in polynomial time. Mahabadi and Trajanovski [28] also proposed coreset-based algorithms for max-sum and max-min diversification under group fairness constraints. However, the above algorithms only consider group fairness (or general matroid) constraints and cannot be directly used in the individually fair variants of diversity maximization.

**Individual Fairness in Clustering Problems.**   Clustering has been studied extensively from a fairness perspective over the past few years. Most previous results consider clustering problems under the notion of *group fairness* [1, 10, 14, 23]. Jung et al. [21] first proposed the notion of *individual fairness* we use in this paper for facility location problems. Following this seminal work, individually fair clustering problems have attracted some attention in recent years. Mahabadi and Vakilian [29] studied center-based clustering problems, such as $k$-median, $k$-means, and $k$-center, with individual fairness constraints. Negahbani and Chakrabarty [35] explored individually fair $k$-clustering with general $\ell_p$-norm objectives using a linear programming approach. Vakilian and Yalçiner [44] studied the problem of $\alpha$-fair $k$-clustering with $l_p$-norm objectives, achieving improved approximation factors in both fairness and cost. Han et al. [17] considered individually fair $k$-center with outliers and proposed a 4-approximation algorithm. Bateni et al. [5] designed a fast local-search algorithm for individually fair $k$-means clustering with improved time complexity. These approaches provide inspiration for the algorithms presented in this paper. However, they cannot be applied directly to diversity maximization, as they are originally designed for clustering problems.

## 3   Problem Definition

Let $(P, d)$ denote a metric space, where $P$ is a set of $n$ points and $d : P \times P \to \mathbb{R}_{\geq 0}$ is a distance function that measures the dissimilarity between any pair of points in $P$ and satisfies the axioms of (i) *identity of indiscernibles*, (ii) *symmetry*, and (iii) *triangle inequality*. We use $S \subseteq P$ to denote a subset of points. The number of points to select, unless otherwise specified, is denoted by $k \in \mathbb{Z}_+$.

Then, we define the objective functions of the max-min, max-sum, and sum-min diversification as follows. They are all distance-based objectives; that is, they are formulated in terms of the pairwise distances between the chosen points. Let $d(u, v)$ be the distance between $u$ and $v$, and for a set of points $T$, let $d(u, T) = \min_{v \in T} d(u, v)$. The diversity functions for the three objectives are as follows:

1. **Max-Min Diversification:** $\mathsf{div}_{\mathrm{mm}}(S, P) = \min_{u \in S} d(u, S \backslash \{u\})$;
2. **Max-Sum Diversification:** $\mathsf{div}_{\mathrm{ms}}(S, P) = \sum_{u \in S} \sum_{v \in S \backslash \{u\}} d(u, v)$;
3. **Sum-Min Diversification:** $\mathsf{div}_{\mathrm{sm}}(S, P) = \sum_{u \in S} d(u, S \backslash \{u\})$.

Max-min diversification aims to maximize the minimum distance between any pair of distinct points in the selected set $S$. Intuitively, it encourages the selected points to be as far apart as possible, ensuring separation among all points. Max-sum diversification aims to maximize the sum of all pairwise distances among the points in the set $S$. Thus, it favors a set in which the points are collectively dispersed across the space. Sum-min diversification aims to maximize the sum of the distances from all points $u \in S$ to their nearest neighbor within $S \backslash \{u\}$. It strikes a balance between local and global diversity, promoting selections where each point is relatively well-separated from at least one close neighbor, without maximizing all pairwise distances.

Next, we introduce some notation to formally define the notion of individual fairness we consider in this paper. For every point $v \in P$, we use $B(v, r) := \{u \in P : d(v, u) \leq r\}$ to denote the subset of all points in $P$ that are at distance at most $r$ from $v$ and call it the ball of radius $r$ centered at $v$.

**Definition 1** (Fair Radius). *Let $l \in [n]$ be a fairness parameter. For every point $v \in P$, we define the fair radius $r_l(v)$ as the minimum distance $r$ such that $|B(v, r)| \geq \frac{n}{l}$. When $l = k$, we drop the subscript for simplicity and use $r(\cdot)$ to denote $r_k(\cdot)$.*

Then, we formally define the notion of $\alpha$-fair $k$-selection [44], a variant of the individual fairness notion from [21, 29].

**Definition 2** ($\alpha$-Fair $k$-Selection). *A set of $k$ points $S \subseteq P$ is $\alpha$-fair if for every point $x \in P$, $d(x, S) \leq \alpha r_k(x)$.*

According to [21], there always exists a feasible solution when $\alpha \geq 2$. Ideally, we would like to find a solution with $\alpha = 1$, which would fully satisfy the original definition of individual fairness [21, 29]. However, since deciding whether a given set of points $P$ admits a fair selection with $\alpha = 1$ is NP-hard [21], it is unlikely that such solutions can be found in polynomial time unless P = NP. Consequently, we aim to provide a bicriteria approximation guarantee instead.

**Definition 3** (Bicriteria Approximation). *An algorithm is a $(\beta, \gamma)$-bicriteria approximation for $\alpha$-fair $k$-selection w.r.t. a given diversity function if for any set of points $P$ the solution $\mathrm{SOL}$ returned by the algorithm on $P$ satisfies the following properties:*

1. *$\mathrm{div}(\mathrm{OPT}, P) \leq \beta \cdot \mathrm{div}(\mathrm{SOL}, P)$, where $\mathrm{OPT}$ denotes the optimal set of $k$ points for $\alpha$-fair $k$-selection of $P$ w.r.t. the given diversity function. In particular, $\mathrm{div}(\mathrm{OPT}, P) = 0$ if an $\alpha$-fair $k$-selection does not exist for $P$.*
2. *$\mathrm{SOL}$ is a $(\gamma \cdot \alpha)$-fair $k$-selection of $P$.*

**Hardness of Approximation.** Ravi et al. [39] showed that, unless P = NP, there does not exist any polynomial-time $\beta$-approximation algorithm for the unconstrained max-min diversification problem when $\beta < 2$. Bhaskara et al. [6] further proved that, under the planted clique conjecture, there exists no polynomial-time $\beta$-approximation algorithm for the unconstrained max-sum and sum-min diversification problems when $\beta < 2$. Since the unconstrained setting can be viewed as the special case of our individually fair setting with $\alpha = \infty$, the hardness results in the unconstrained case naturally generalize to our problems, as stated below.

**Theorem 1.** *There exists no polynomial-time $\beta$-approximation algorithm for individually fair max-min diversification with $\beta < 2$, unless P = NP.*

**Theorem 2.** *There exists no polynomial-time $\beta$-approximation algorithm for individually fair max-sum and sum-min diversification with $\beta < 2$ under the planted clique conjecture.*

---

**Algorithm 1:** IFRGENERATE

---

**Input:** Fairness parameter $\alpha$
**Output:** A set of individual fairness regions for given parameters $\alpha, k$
**Initialize** the set of covered points $Z \leftarrow \emptyset$ and the set of centers of selected balls $\mathcal{C} \leftarrow \emptyset$
**while** $Z \neq P$ **do**
    $c \leftarrow \arg\min_{x \in P \setminus Z} r(x)$
    $\mathcal{C} \leftarrow \mathcal{C} \cup \{c\}$
    $Z \leftarrow Z \cup \{x \in P \setminus Z | d(x, c) \leq 2\alpha \cdot r(x)\}$
**end while**
**return** $\{B(c, \alpha \cdot r(c)) : c \in \mathcal{C}\}$

---

In previous studies on individually fair clustering [29, 44], connections between individual fairness and partition matroid constraints have been established. We now introduce the concept of matroid constraints so that our subsequent individually fair algorithms can be framed as solutions rooted in them. Note that the group fairness constraint has also been shown to be a type of matroid constraint.

**Definition 4** (Matroid Constraint). *A matroid $\mathcal{M}$ is defined as a family of subsets of the ground set of points $\mathcal{E}(\mathcal{M}) = P$, called independent sets. The set of independent sets $S$ of a matroid $\mathcal{M}$ is denoted by $\mathcal{I}(\mathcal{M})$. For a given matroid $\mathcal{M}$, the associated matroid constraint is $S \in \mathcal{I}(\mathcal{M})$. As is standard, $\mathcal{M}$ is a uniform matroid of rank $r$ if $\mathcal{I}(\mathcal{M}) := \{X \subseteq \mathcal{E}(\mathcal{M}) : |X| \leq r\}$. A partition matroid is the direct sum of uniform matroids. Note that uniform matroid constraints are equivalent to cardinality constraints, i.e., $|S| \leq k$. This definition follows [43].*

To satisfy individual fairness constraints, we introduce a special structure called *individual fairness regions*, which partitions the points based on their distances in the metric space. This notion is useful in our algorithms for connecting individual fairness to matroid constraints.

**Definition 5** (Individual Fairness Region). *A set of balls $\mathcal{B} = \{B(c_1, \alpha \cdot r(c_1)), \ldots, B(c_m, \alpha \cdot r(c_m))\}$, where $m \leq k$, is called a set of individual fairness regions if it satisfies the following properties:*

1. *For every $x \in P : d(x, \{c_1, \ldots, c_m\}) \leq 2\alpha \cdot r(x)$;*

2. *For any pair of centers $c_i, c_j, d(c_i, c_j) > 2\alpha \cdot \max\{r(c_i), r(c_j)\}$; in other words, individual fairness regions are disjoint from each other.*

## 4 Our Algorithms

In this section, we reduce our individually fair diversity maximization problem to data selection under partition matroid constraints. We first present an algorithm, IFRGENERATE, that, given a set of points $P$ and a fairness parameter $\alpha$, returns a set of individual fairness regions.

**Overview of Algorithm 1.** IFRGENERATE is designed to address the problem of maximizing diversity under individual fairness constraints by reducing it to the problem under partition matroid constraints. Similar algorithms have been used for individually fair clustering, and they are crucial for satisfying individual fairness constraints in clustering problems [29, 44]. The algorithm takes as input a fairness parameter $\alpha$, the size of the input data set $n$, and the desired size of the output set $k$. The procedure begins by initializing two empty sets: a set of covered points $Z \leftarrow \emptyset$ and a set of selected ball centers $\mathcal{C} \leftarrow \emptyset$. It then enters a loop that continues as long as the set of uncovered points $P \setminus Z$ is non-empty. At each iteration, the algorithm selects a center $c$ that has the minimal fair radius in the uncovered points, adds $c$ to the set of centers $\mathcal{C}$, and updates the set of covered points $Z$ by including all points $x \in P \setminus Z$ whose distance $d(x, c)$ is at most $2\alpha \cdot r(x)$. This process repeats until all points are covered. The output of the algorithm is a set of balls $\{B(c, \alpha r(c)) : c \in \mathcal{C}\}$, where each ball should consist of a selected center $c$ and its radius $\alpha r(c)$. This set of balls can be treated as different groups, connecting our problem with that under the partition matroid constraints.

We now prove that IFRGENERATE generates individual fairness regions as defined in Definition 5.

**Lemma 1.** *Let $k$ be a positive integer and $\alpha$ be a parameter that indicates the desired fairness guarantee. Algorithm 1 returns a set of at most $k$ individual fairness regions in $O(nk)$ time.*

---
**Algorithm 2:** OMSGENERATE
---
**Input:** Set of points $P$, desired number of selected points $k$, fairness parameter $\alpha$

**Output:** Individual fairness regions and $k_i$ in the original metric space

**Compute** a set of individual fairness regions $\mathcal{B} = \{B_1, \cdots, B_m\}$ on $(P, k, \alpha)$ using Algorithm 1

**Let** $P_0 = \{v_0 | v \in P \backslash (B_1 \cup \cdots \cup B_m)\}$ be the points not in individual fairness regions

$k_i = k - m + 1$ **for all** $i \in [m]$ ▷ *{denote that we pick at most $k - m + 1$ centers from each individual fairness ball}*

$k_0 = k$ ▷ *{denote that we pick at most $k$ centers from $P_0$}*

**return** $\{(P_0, k_0), (B_1, k_1), \cdots, (B_m, k_m)\}$
---

See the proof in Appendix B.1. Similar to individually fair clustering [29, 44], the benefit of a set of individual fairness regions is that it reduces the problem of finding an $\alpha$-fair selection to a data selection problem with lower bound requirements, i.e., at least one point must be selected from each individual fairness region. Lemma 2 indicates that a set of points $S$ is *feasible* w.r.t. a set of individual fairness regions $\mathcal{B}$, if for every ball $B \in \mathcal{B}, |S \cap B| > 0$.

**Lemma 2.** *Let $\mathcal{B} = \{B(c_1, \alpha \cdot r(c_1)), \ldots, B(c_m, \alpha \cdot r(c_m))\}$ be a set of individual fairness regions obtained from Algorithm 1 for a set of points $P$ with parameters $k$ and $\alpha$. Then, any set of points $S$ that is feasible w.r.t. $\mathcal{B}$ is $(3\alpha)$-fair.*

See the proof in Appendix B.2. Based on the lemmas above, we now introduce our algorithm for maximizing diversity under individual fairness constraints. We first construct individual fairness regions on the original data $P$ using Algorithm 2.

**Overview of Algorithm 2.** OMSGENERATE is designed to address the problem of selecting a diverse subset of points while satisfying the fairness constraints in the given dataset $P$, specifically by generating a set of individual fairness regions in $P$. The algorithm takes as input a set of points $P$, a desired number of selected points $k$, a fairness parameter $\alpha$, and the original metric space. It begins by computing a set of individual fairness regions $\mathcal{B} = \{B_1, \ldots, B_m\}$ using Algorithm 1 (IFRGENERATE) on the input $(P, k, \alpha)$. Next, it identifies the set of points $P_0$ that are not covered by these fairness regions, i.e., points outside the union of all fairness regions in $\mathcal{B}$. For each individual fairness region $B_i$, the algorithm sets the maximum number of points that can be selected from $B_i$ at $k_i = k - m + 1$. In addition, it sets $k_0 = k$, which is the maximum number of centers that can be selected from the uncovered points $P_0$. Note that $k_0, k_1, \ldots, k_m$ do not denote the number of points that should be selected but the maximum number that can be selected in a region. In practical implementations, the number of points to be selected may differ from $k_0, k_1, \ldots, k_m$, as they are only the upper bounds on the number of points to select. Our ultimate goal is always to select $k$ points as the final result. Finally, the algorithm returns a set of individual fairness regions $\{(P_0, k_0), (B_1, k_1), \ldots, (B_m, k_m)\}$, representing the individual fairness regions in $P$ and the maximum number of points that can be selected from each region. Since the computation can be mainly attributed to Algorithm 1, the time complexity of Algorithm 2 is $O(nk)$ as well.

Now, we demonstrate that with a $\beta$-approximation algorithm for different diversity maximization problems under partition matroid constraints, a bicriteria approximation for $\alpha$-fair $k$-selection exists. We first give the approximation for max-min diversification under individual fairness constraints.

**Theorem 3.** *Suppose that there exists a $\beta$-approximation algorithm for max-min diversification under partition matroid constraints. Then, there exists a $(\beta, 3)$-bicriteria approximation for $\alpha$-fair $k$-selection in max-min diversification.*

*Proof Sketch.* We show that the solution from a $\beta$-approximation algorithm (MAXMINALG) for max-min diversification under partition matroid constraints is a $(\beta, 3)$-bicriteria approximation for $\alpha$-fair $k$-selection on metric space $P$. The solution $\text{SOL}_G$ from MAXMINALG satisfies the partition matroid constraints, ensuring at least one point per fairness region $B_i$. By Lemma 2, this implies that $\text{SOL}_G$ is $(3\alpha)$-fair. We show that the diversity of the optimal solution $\text{OPT}_G$ on $P$ is equal to that of $\text{OPT}'_G$ on a constructed instance $P'$. By mapping the solutions between $P$ and $P'$, we show $\text{div}_{\text{mm}}(\text{OPT}_G, P) = \text{div}_{\text{mm}}(\text{OPT}'_G, P')$.

For the $\alpha$-fair optimal solution $\text{OPT}_I$ on $P$, we can construct a feasible solution $\text{OPT}'_C$ on $P'$ with equal diversity while satisfying the partition matroid constraints. Since $\text{OPT}'_G$ is opti-

mal on $P'$, $\mathsf{div}_{\mathrm{mm}}(\mathrm{OPT}'_G, P') \geq \mathsf{div}_{\mathrm{mm}}(\mathrm{OPT}'_C, P') = \mathsf{div}_{\mathrm{mm}}(\mathrm{OPT}_I, P)$. Given that MAXMI-NALG is $\beta$-approximate, $\mathsf{div}_{\mathrm{mm}}(\mathrm{OPT}_G, P) \leq \beta \cdot \mathsf{div}_{\mathrm{mm}}(\mathrm{SOL}_G, P)$, thus $\mathsf{div}_{\mathrm{mm}}(\mathrm{OPT}_I, P) \leq \beta \cdot \mathsf{div}_{\mathrm{mm}}(\mathrm{SOL}_G, P)$. Therefore, $\mathrm{SOL}_G$ is a $(\beta, 3)$-bicriteria approximation. $\square$

See the full proof in Appendix B.4.

After proving the existence of a bicriteria approximation algorithm for $\alpha$-fair $k$-selection with a max-min objective, we then prove that the algorithm for the max-sum objective also exists.

**Theorem 4.** *Suppose that there exists a $\beta$-approximation algorithm for max-sum diversification under partition matroid constraints. Then, there exists a $(\beta(4 + \varepsilon), 3)$-bicriteria approximation for $\alpha$-fair $k$-selection in max-sum diversification.*

*Proof Sketch.* We show that the solution from a $\beta$-approximation algorithm (MAXSUMALG) for max-sum diversification under partition matroid constraints is a $(\beta(4+\varepsilon), 3)$-bicriteria approximation for $\alpha$-fair $k$-selection on metric space $P$. We establish that $\mathsf{div}_{\mathrm{ms}}(\mathrm{OPT}_G, P) \geq \frac{1}{4+\varepsilon}\mathsf{div}_{\mathrm{ms}}(\mathrm{OPT}'_G, P')$. When $\mathsf{div}_{\mathrm{ms}}(\mathrm{OPT}_G, P) < \mathsf{div}_{\mathrm{ms}}(\mathrm{OPT}'_G, P')$, $\mathrm{OPT}'_G$ may include multiple copies of points in $P$. In the extreme case, $k$ points in $P'$ map to $\lfloor \frac{k}{2} \rfloor$ points in $P$, and we analyze basic cases (e.g., four points in $P'$ from two in $P$) to show $\mathsf{div}_{\mathrm{ms}}(\mathrm{OPT}'_G, P') \leq (4+\varepsilon) \cdot \mathsf{div}_{\mathrm{ms}}(\mathrm{ORI}, P)$, where ORI are distinct points in $P$. Since $\mathsf{div}_{\mathrm{ms}}(\mathrm{OPT}_G, P) \geq \mathsf{div}_{\mathrm{ms}}(\mathrm{ORI}, P)$, we get $\mathsf{div}_{\mathrm{ms}}(\mathrm{OPT}_G, P) \geq \frac{1}{4+\varepsilon}\mathsf{div}_{\mathrm{ms}}(\mathrm{OPT}'_G, P')$.

For the $\alpha$-fair optimal solution $\mathrm{OPT}_I$ on $P$, we construct $\mathrm{OPT}'_C$ on $P'$ with equal diversity while satisfying the partition matroid constraints. Since $\mathrm{OPT}'_G$ is optimal on $P'$, $\mathsf{div}_{\mathrm{ms}}(\mathrm{OPT}'_G, P') \geq \mathsf{div}_{\mathrm{ms}}(\mathrm{OPT}'_C, P') = \mathsf{div}_{\mathrm{ms}}(\mathrm{OPT}_I, P)$. In combination with the $\beta$-approximation of MAXSUMALG, $\mathsf{div}_{\mathrm{ms}}(\mathrm{OPT}_I, P) \leq (4 + \varepsilon) \cdot \mathsf{div}_{\mathrm{ms}}(\mathrm{OPT}_G, P) \leq \beta(4 + \varepsilon) \cdot \mathsf{div}_{\mathrm{ms}}(\mathrm{SOL}_G, P)$. Therefore, $\mathrm{SOL}_G$ is a $(\beta(4 + \varepsilon), 3)$-bicriteria approximation. $\square$

See the full proof in Appendix B.5.

We then consider the same circumstance for sum-min diversification.

**Theorem 5.** *Suppose that there exists a $\beta$-approximation algorithm for sum-min diversification under partition matroid constraints with $1 < k < n/3$. Then, there exists a $(\beta(4 + \varepsilon), 3)$-bicriteria approximation for $\alpha$-fair $k$-selection in sum-min diversification.*

*Proof Sketch.* We show that the solution from a $\beta$-approximation algorithm (SUMMINALG) for $k$-selection under partition matroid constraints for sum-min diversification is a $(\beta(4 + \varepsilon), 3)$-bicriteria approximation for $\alpha$-fair $k$-selection on metric space $P$. We establish that $\mathsf{div}_{\mathrm{sm}}(\mathrm{OPT}_G, P) \geq \frac{1}{4+\varepsilon}\mathsf{div}_{\mathrm{sm}}(\mathrm{OPT}'_G, P')$. When $\mathsf{div}_{\mathrm{sm}}(\mathrm{OPT}_G, P) < \mathsf{div}_{\mathrm{sm}}(\mathrm{OPT}'_G, P')$, $\mathrm{OPT}'_G$ includes multiple copies of the points in $P$. With $z \leq k$ distinct points in $P$ corresponding to $\mathrm{OPT}'_G$, pairs of copies contribute $2(k - z)\varepsilon\delta$ to $\mathsf{div}_{\mathrm{sm}}(\mathrm{OPT}'_G, P')$. Defining $D = \mathsf{div}_{\mathrm{sm}}(\mathrm{OPT}'_G, P') - 2(k - z)\varepsilon\delta$, we show $D \leq \mathsf{div}_{\mathrm{sm}}(\mathrm{OPT}^z_G, P)$, where $\mathrm{OPT}^z_G$ is the optimal $z$-point solution in $P$. By Lemma 4 (see Appendix B.6), $\mathsf{div}_{\mathrm{sm}}(\mathrm{OPT}^z_G, P) \leq 4 \cdot \mathsf{div}_{\mathrm{sm}}(\mathrm{OPT}_G, P)$ for $1 < k < n/3$. Since $2(k - z)\varepsilon\delta \leq \varepsilon \cdot \mathsf{div}_{\mathrm{sm}}(\mathrm{OPT}_G, P)$, we get $\mathsf{div}_{\mathrm{sm}}(\mathrm{OPT}'_G, P') \leq (4 + \varepsilon) \cdot \mathsf{div}_{\mathrm{sm}}(\mathrm{OPT}_G, P)$.

For the $\alpha$-fair optimal solution $\mathrm{OPT}_I$ on $P$, we construct $\mathrm{OPT}'_C$ on $P'$ with equal diversity satisfying partition matroid constraints. Since $\mathrm{OPT}'_G$ is optimal on $P'$, $\mathsf{div}_{\mathrm{sm}}(\mathrm{OPT}'_G, P') \geq \mathsf{div}_{\mathrm{sm}}(\mathrm{OPT}'_C, P') = \mathsf{div}_{\mathrm{sm}}(\mathrm{OPT}_I, P)$. Combining with the $\beta$-approximation of SUMMINALG, $\mathsf{div}_{\mathrm{sm}}(\mathrm{OPT}_I, P) \leq (4 + \varepsilon) \cdot \mathsf{div}_{\mathrm{sm}}(\mathrm{OPT}_G, P) \leq \beta(4 + \varepsilon) \cdot \mathsf{div}_{\mathrm{sm}}(\mathrm{SOL}_G, P)$. Therefore, $\mathrm{SOL}_G$ is a $(\beta(4 + \varepsilon), 3)$-bicriteria approximation. $\square$

See the full proof in Appendix B.6.

According to Theorems 3–5, we have the following theorems with specific approximation factors. Note that the computation of $\varepsilon$ differs between these theorems; detailed explanations are provided in Appendix C.

**Theorem 6.** *For any $\alpha \geq 1$, there exists a $O(mkn + m^k \log \frac{1}{\varepsilon})$ time algorithm that computes a $(5 + \varepsilon, 3)$-bicriteria approximate solution to the $\alpha$-fair $k$-selection of max-min diversification under individual fairness constraints.*

The proof follows from Theorem 3 and the $(5 + \varepsilon)$-approximation algorithm from [46] for max-min diversification under partition matroid constraints. We note that the time complexity of this algorithm is not linear w.r.t. $k$. More detailed discussions are provided in Appendix D.

**Theorem 7.** *For any $\alpha \geq 1$, there exists a polynomial time algorithm that computes a $(8 + \varepsilon, 3)$-bicriteria approximate solution to the $\alpha$-fair $k$-selection of max-sum diversification under individual fairness constraints.*

The proof follows from Theorem 4 and the 2-approximation algorithm in [2] for max-sum diversification under partition matroid constraints.

**Theorem 8.** *For any $\alpha \geq 1$, there exists a nearly linear-time algorithm that computes a $(32 + \varepsilon, 3)$-bicriteria approximate solution to the $\alpha$-fair $k$-selection of sum-min diversification under individual fairness constraints.*

The proof follows from Theorem 5 and the 8-approximation algorithm in [6] for sum-min diversification under partition matroid constraints.

## 5 Experiments

In this section, we empirically compare our algorithms with unconstrained diversity maximization algorithms. The experiments focus on presenting the trade-off between fairness and diversity loss of our algorithms.

**Implementation.** All experiments were carried out on a server with an Intel(R) Xeon(R) Gold 6134 CPU @3.20GHz (2 processors) and 128GB RAM running Windows Server 2019 Datacenter. The algorithms were implemented in Python 3. Our code and data are publicly available at `https://github.com/HonokaKousaka/IFDM`.

**Data Sets.** In the experiments, we used three public real-world data sets and one synthetic data set. The basic information for each data set is shown in Table 1. For MovieLens, the user vectors are obtained through matrix factorization using LIBMF [11]. We randomly sampled 1,000 points from each data set for evaluation.

Table 1: Statistics of data sets in the experiments, where $n$ is the number of data points and $dim$ is the dimensionality.

| Dataset | Description | $n$ | $dim$ |
|---|---|---|---|
| CelebA [26] | Features for celebrity images extracted by VGG16 | 202,599 | 25,088 |
| GloVe [37] | Global vectors for word representation | 400,000 | 100 |
| MovieLens [18] | User vectors obtained from the rating matrix | 162,541 | 50 |
| Gaussian | Gaussian blobs by `make_blobs` in scikit-learn [36] | 1,000,000 | 20 |

**Experimental Setup.** For max-min diversification under individual fairness constraints, we implemented FMMD-S [46], a $(5 + \varepsilon)$-approximation algorithm for max-min diversification under partition matroid constraints in $O(mkn + m^k \log \frac{1}{\varepsilon})$ time, and fixed $\varepsilon = 0.05$ for FMMD-S. We compared our algorithm with GMM [16], which provides a 2-approximation for unconstrained max-min diversification. For max-sum diversification under individual fairness constraints, we implemented the local search algorithm in [2], which is a 2-approximation algorithm for max-sum diversification under general matroid constraints in $O(\frac{n}{\varepsilon} \log(k))$ time, and set $\varepsilon = 0.05$ in the algorithm. We compared our algorithm with the greedy algorithm in [19], which guarantees a 2-approximation for unconstrained max-sum diversification. For sum-min diversification under individual fairness constraints, we implemented the coreset-based algorithm in [28], which provides an $O(m^2 \cdot \log k)$-approximation for sum-min diversification under group fairness constraints in polynomial time. To obtain an unconstrained solution for sum-min diversification, we placed all points in an individual fairness region and ran our algorithm accordingly. We fixed $\alpha = 1$ in all experiments so that each algorithm makes the best effort to ensure individual fairness. We implemented all algorithms in Python 3 and used the Gurobi optimizer to solve ILPs in FMMD-S. To compare with unconstrained optimal solutions for quantifying exact utility losses, we also ran the Gurobi optimizer to solve the ILPs for max-min, max-sum, and sum-min diversification within a 30-minute time limit.

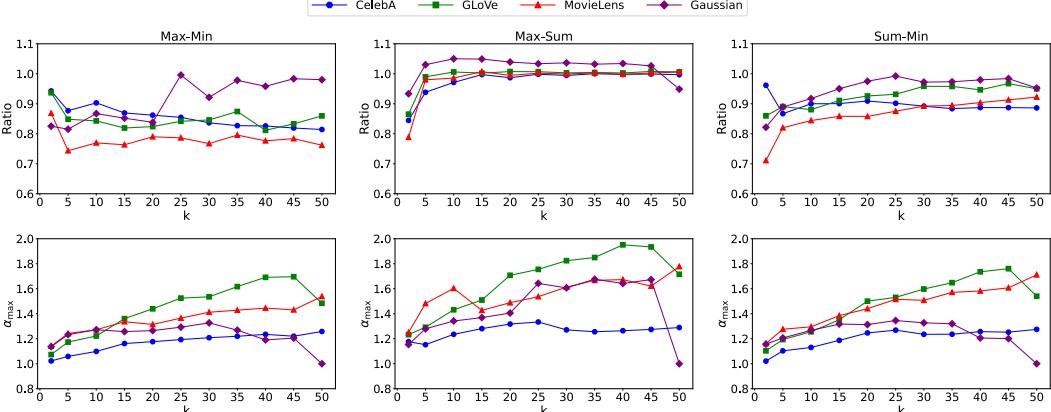

Figure 1: Overall experimental results. The first row illustrates the ratios of the diversity values of the solutions returned by our algorithms over the diversity values of the best solutions returned by unconstrained algorithms when $k = 2, 5, 10, \ldots, 50$ and $\alpha = 1$; the second row presents the $\alpha_{\max}$ values of the solutions returned by our algorithms.

**Evaluation Measures.** In terms of individual fairness, we define $\alpha_{\max} = \max_{x \in P} d(x, \mathrm{SOL})/r(x)$ as the performance metric. Specifically, $\alpha_{\max}$ indicates how close a solution is to satisfying the individual fairness constraint on $P$: The smaller $\alpha_{\max}$, the fairer the solution SOL. If $\alpha_{\max} \leq 1$, the solution SOL strictly satisfies the individual fairness constraint. In terms of utility, we compute the ratio of the diversity value of a fair solution to that of an unconstrained solution returned by an approximation or an exact ILP-based algorithm, using this ratio as the performance metric. The larger the ratio, the higher the utility of the solution. When using approximation algorithms to obtain unconstrained solutions, we ran them 10 times and selected the solution with the largest diversity value. We also ran our proposed algorithms ten times and used the average of both measures for evaluation.

**Experimental Results.** In Figure 1, we present the experimental results for three diversity objectives across four data sets. We observe that our algorithms consistently provide approximate solutions with diversity losses of no more than 30% compared to unconstrained (approximate) solutions, while always guaranteeing that the value of $\alpha_{\max}$ is below 2. In Table 2, we also present the ratios of the diversity values of our solutions to the optimal diversity values computed from the ILPs solved by Gurobi. As shown, our algorithms limit utility losses relative to unconstrained (optimal) solutions to at most 35%, further validating their effectiveness in ensuring diversity while satisfying individual fairness constraints. These results demonstrate that our algorithms strike an effective balance between individual fairness and diversity.

Table 2: Ratios of the diversity values of the solutions of our proposed algorithms over the diversity values of the optimal unconstrained solutions by ILPs when $k = 5, 10, 20$ and $\alpha = 1$.

| Dataset | Max-Min | | | Max-Sum | | | Sum-Min | | |
|---|---|---|---|---|---|---|---|---|---|
| | $k = 5$ | $k = 10$ | $k = 20$ | $k = 5$ | $k = 10$ | $k = 20$ | $k = 5$ | $k = 10$ | $k = 20$ |
| CelebA | 0.827 | 0.807 | 0.794 | 0.923 | 0.967 | 0.985 | 0.703 | 0.769 | 0.842 |
| GloVe | 0.780 | 0.793 | 0.824 | 0.947 | 0.980 | 0.991 | 0.754 | 0.835 | 0.910 |
| MovieLens | 0.701 | 0.724 | 0.764 | 0.925 | 0.961 | 0.977 | 0.676 | 0.757 | 0.830 |
| Gaussian | 0.819 | 0.845 | 0.819 | 0.968 | 0.988 | 0.996 | 0.831 | 0.893 | 0.955 |

Figure 2 shows the running times of our algorithms on each data set. In terms of time efficiency, taking CelebA as an example, the process of computing an individually fair solution (including Algorithms 1 & 2, and the diversification algorithms with matroid constraints) takes no more than 4 seconds when $k \leq 50$ for max-min and sum-min diversification. For max-sum diversification, because $\varepsilon$ is very small, the local search requires many iterations to meet the stop condition; thus, it takes about 300 seconds when $k = 50$.

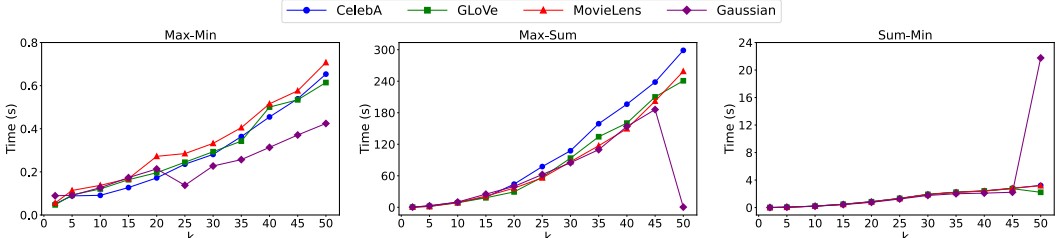

Figure 2: Running time (in seconds) of our algorithms when $k = 2, 5, 10, \ldots, 50$ and $\alpha = 1$.

We observe that when $k = 50$, the synthetic Gaussian data set exhibits anomalous results in the ratios, the $\alpha_{\max}$ values, and the running time for max-sum and sum-min diversification. This phenomenon can be attributed to the data generation process, since the Gaussian data set is constructed by sampling points around 50 Gaussian centers. As a result, when $k = 50$, the data set naturally forms exactly 50 individual fairness regions, each containing 20 points. Consequently, every data point falls into one of these regions, and selecting one point from each region is sufficient to achieve high diversity while guaranteeing individual fairness. This leads to $\alpha_{\max} \leq 1$ when $k = 50$; that is, the proposed algorithms provide solutions that strictly satisfy the individual fairness constraints. Moreover, for max-sum diversification, the local search procedure requires only a few iterations before meeting the stop condition because the initial solution already contains one point from each region. In contrast, for sum-min diversification, our proposed algorithm uses GMM to fit within each of the 50 fairness regions and determine potential candidates. Consequently, performing 50 independent GMM instances significantly increases the runtime when $k = 50$.

## 6 Conclusion

In this paper, we study the diversity maximization problem under individual fairness constraints, namely $\alpha$-fair $k$-selection. By generating individual fairness regions to partition data points and utilizing existing approximation algorithms for diversity maximization under matroid constraints, we propose a $(5 + \varepsilon, 3)$-bicriteria approximation algorithm for max-min diversification, $(8 + \varepsilon, 3)$-bicriteria approximation algorithm for max-sum diversification, and $(32 + \varepsilon, 3)$-bicriteria approximation algorithm for sum-min diversification for any $\alpha \geq 1$ in any metric space. Experimental results demonstrate that our proposed algorithms efficiently provide individually fair, highly diverse subsets on real-world and synthetic datasets.

There are still some open problems for future work. Given that the approximation factors of our algorithms are determined by those for diversity maximization under matroid constraints, a possible direction is to improve the approximation ratio for individually fair diversity maximization by utilizing better matroid-constrained diversity maximization algorithms. In addition, we acknowledge that we have not proved whether individually fair diversity maximization problems are strictly harder to approximate than their unconstrained counterparts. Establishing such hardness results remains open and would shed light on further improvements in approximation factors. Another promising avenue for exploration is to extend the problems and proposed algorithms to streaming, distributed, and deletion-robust settings.

## Acknowledgments and Disclosure of Funding

This work was supported by the National Natural Science Foundation of China (Grant No. 62202169).

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

# A  Algorithm 3 (PSGENERATE) (Used for Proof Only)

Consider an instance of $\alpha$-fair $k$-selection on a point set $P$. Let $\mathcal{B}$ be the set of individual fairness regions of $P$ with parameters $k$ and $\alpha$ constructed using Algorithm 2. Then, given an instance of $\alpha$-fair $k$-selection, Algorithm 3 builds an instance of $k$-selection under partition matroid constraints.

**Overview of Algorithm 3.** Existing studies on individually fair clustering [44] have implemented a similar instance to generate an individually fair solution for clustering problems. Our implementation of Algorithm 3 is a little different, since the instance generated is not directly involved in our solution generation process. However, it serves a critical role in our theoretical analysis. Unlike prior work, where the constructed instance is typically used as a direct component in producing the final clustering solution, our approach leverages the instance primarily to facilitate the proof of our theorem guarantees, ensuring that the diversity properties hold under the individual fairness constraints. PSGENERATE takes as input a point set $P$, the desired number of selected points $k$, a fairness parameter $\alpha$, and an accuracy parameter $\varepsilon < 1/2$. It begins by computing a set of individual fairness regions $\mathcal{B} = B_1, \ldots, B_m$ using Algorithm 2 on $(P, k, \alpha)$. Next, it creates copies of the original point set $P$ and each individual fairness region $B_i$, denoted as $\overline{P}_0$ and $\overline{B}_i$ respectively, and constructs an extended set $P'$ by including $\overline{P}_0$ and $\overline{B}_i$. Note that $P'$ has two distinct copies of the points that belong to an individual fairness ball of $\mathcal{B}$. A modified distance function $d'$ is then defined, where $d'(u, u) = 0$ for all $u \in P'$, $d'(v_x, u_y) = d(v, u)$ for distinct $v, u \in P'$ with $v \neq u$, and $d'(v_x, v_y) = \varepsilon \delta$ for $v_x, v_y \in P'$, where $\delta \leftarrow \min_{x,y \in P} d(x, y)$. The algorithm finally returns the tuple $(P', \{(\overline{P}_0, \overline{k}_0), (\overline{B}_1, \overline{k}_1), \ldots, (\overline{B}_m, \overline{k}_m)\}, d')$, providing an instance corresponding to the given $\alpha$-fair $k$-selection problem for the proof purpose. Given that the individual fairness regions in $P$ have been computed, the time complexity of Algorithm 3 is $O(n)$, since most of the steps only require iterating over all points in $P$. Note that we do not need to implement Algorithm 3 in practice.

---

**Algorithm 3:** PSGENERATE

**Input:** set of points $P$, desired number of selected points $k$, fairness parameter $\alpha$, accuracy parameter $\varepsilon < 1/2$

**Output:** an instance of $k$-selection under partition matroid constraints w.r.t. the given instance of $\alpha$-fair $k$-selection for the proof

**Compute** the set $\{(P_0, k_0), (B_1, k_1), \cdots, (B_m, k_m)\}$ on $(P, k, \alpha)$ using Algorithm 2

**Let** $\overline{P}_0 = \{v_0 | v \in P\}$ be a copy of $P$

**Let** $\overline{B}_i = \{v_i | v \in B_i\}$ be a copy of $B_i$ for all $B_i \in \mathcal{B}$

$P' \leftarrow \overline{P}_0 \cup \left(\bigcup_{B_i \in \mathcal{B}} \overline{B}_i\right)$ ▷ $\{P'$ has two distinct copies of the points that belong to an individual fairness ball of $\mathcal{B}.\}$

▷ Construct a distance function $d' : P' \times P' \rightarrow \mathbb{R}^+$

$\overline{k}_0 = k_0$ ▷ $\{$denotes that we pick at most $k_0$ points from $\overline{P}_0\}$

$\overline{k}_i = k_i$ **for all** $i \in [m]$ ▷ $\{$denotes that we pick at most $k_i$ points from $\overline{B}_i\}$

**Let** $\delta \leftarrow \min_{x,y \in P} d(x, y)$

**Let** $d'(u, u) = 0$ **for all** $u \in P'$

**Let** $d'(v_x, u_y) = d(v, u)$ **for all** $v_x, u_y \in P'$ where $v \neq u$

**Let** $d'(v_x, v_y) = \varepsilon \delta$ **for all** $v_x, v_y \in P'$

**return** $(P', \{(\overline{P}_0, \overline{k}_0), (\overline{B}_1, \overline{k}_1), \cdots, (\overline{B}_m, \overline{k}_m)\}, d')$

---

We show that the distance function $d'$ in Algorithm 3 is a metric distance.

**Lemma 3.** *The distance function $d' : P' \times P' \rightarrow \mathbb{R}^+$ constructed in Algorithm 3 constitutes a metric space. (See the proof in Appendix B.3.)*

# B  Missing Proofs

## B.1  Proof of Lemma 1

*Proof.* First, we show that the set of centers returned by Algorithm 1 satisfies property (1) of the individual fairness regions. For every point $x \in P$ let $c_x$ denote the first center added to $\mathcal{C}$ such that $d(x, c_x) \leq 2\alpha \cdot r(x)$. Hence, $d(x, \mathcal{C}) \leq d(x, c_x) \leq 2\alpha \cdot r(x)$ where the last inequality follows from

the fact that $c_x$ marks $x$ as covered. Next, consider the iteration of the algorithm in which a center $c$ is added to $\mathcal{C}$. Since $c$ is an uncovered point, its distance to any other center $c'$ that is already in $\mathcal{C}$ is more than $2\alpha \cdot r(c) = 2\alpha \cdot \max\{r(c), r(c')\}$ where the equality follows from the fact that centers are picked in a non-decreasing order of their fair radius. Hence, for any pair of centers in $\mathcal{C}$, property (2) holds. Finally, by property (2), the balls of radius $r(\cdot)$ around the centers present in $\mathcal{C}$ are disjoint. Moreover, according to the definition of fair radius, each of the balls $\{B(c, r(c))\}_{c \in \mathcal{C}}$ contains at least $n/k$ points. Hence, the number of individual fairness regions is at most $k$.

Now we consider the time complexity of Algorithm 1. When generating individual fairness regions, the first step (Line 5) is to iterate over all radii to find the minimal fair radius, whose worst time complexity is $O(n)$. The time complexity of Line 6 is obviously $O(1)$. The final step (Line 7) requires iterating over points in $P \backslash Z$, whose time complexity is $O(n)$. Considering the outer `while` loop runs at most $k$ times, the total time complexity of Lines 5, 6, and 7 is $O(nk)$, $O(k)$, and $O(nk)$, respectively. Thus, the total time complexity for generating individual fairness regions is $O(nk)$.  $\square$

## B.2 Proof of Lemma 2

*Proof.* Let $S$ be a set of cluster centers that is feasible w.r.t. $\mathcal{B}$. For every point $x \in P$, let $c_x$ denote the first center chosen by Algorithm 1 such that $d(x, c_x) \leq 2\alpha \cdot r(x)$. Moreover, let $s_x$ denote the center in $S$ such that $s_x \in B(c_x, \alpha \cdot r(c_x))$. Then, for any point $x \in P$, we have

$$
\begin{aligned}
d(x, s_x) &\leq d(x, c_x) + d(c_x, s_x) \\
&\leq 2\alpha \cdot r(x) + d(c_x, s_x) \\
&\leq 2\alpha \cdot r(x) + \alpha \cdot r(c_x) \\
&\leq 3\alpha \cdot r(x),
\end{aligned}
$$

where the first inequality follows from the triangle inequality, the second inequality follows from the property (1) of individual fairness regions, the third inequality follows since $s_x \in B(c_x, \alpha \cdot r(c_x))$, and the last inequality follows since centers are added in a non-decreasing order of their fair radius in line 5 of Algorithm 1, leading to $r(c_x) \leq r(x)$.  $\square$

## B.3 Proof of Lemma 3

*Proof.* Let $u, v, w \in (\mathcal{F} \cup \mathcal{M})$ be three arbitrary points and let $u_P, v_P, w_P$ be their corresponding points from $P$. First, we prove that $d'(u, v) = 0 \iff u = v$. If $u = v$, the distance $d'(u, v)$ is set to zero by line 10. To show the other direction, if $d'(u, v) = 0$ then the constraint $u = v$ for the assignment in line 10 is satisfied since $d(u_P, v_P) > 0$ for all $u_p \neq v_p$ (line 11) and $d'(u, v) = \varepsilon\delta > 0$ when $u_p = v_p$ and $u \neq v$ (line 12). Secondly, we prove the symmetric property $d'(u, v) = d'(v, u)$. If $d'(u, v) = 0$, then by the first part $u = v$ and therefore $d'(v, u) = 0 = d'(u, v)$. Assume $d'(u, v) > 0$, which implies $u \neq v$. If $u_P \neq v_P$, then by line 11 and the metric properties of $d$, $d'(u, v) = d(u_P, v_P) = d(v_P, u_P) = d'(v, u)$ holds. Lastly, we show that the triangle inequality $d'(u, w) \leq d'(u, v) + d'(v, w)$ holds. If $u = w$ then by the first property, $d'(u, w) = 0$ so the equality holds. Assume $u \neq w$ and consider their corresponding points $u_P, w_P$.

1. If $u_P = w_P$, then $d'(u, w) = \varepsilon\delta$. If $v_P = u_P$, then $d'(u, v) = d'(u, w) = \varepsilon\delta$ and therefore $d'(u, w) \leq d'(u, v) + d'(v, w)$ already holds. If $v_P \neq u_P$, then $d'(u, v) = d(u_P, v_P) \geq \min_{x,y \in P} d(x, y) \geq \varepsilon\delta$. Thus, $d'(u, w) \leq d'(u, v) + d'(v, w)$ holds.

2. If $u_P \neq w_P$, then $d'(u, w) = d(u_P, w_P) \geq \varepsilon\delta$. Note that ($u_P = v_P$ and $v_P = w_P$) can not hold, so consider the remaining three cases:

   (a) $v_P = w_P$ and $u_P \neq v_P$. Then $d'(u, w) = d'(u, v)$ and $d'(u, w) \leq d'(u, v) + d'(v, w)$;
   (b) $u_P = v_P$ and $v_P \neq w_P$. Then $d'(u, w) = d'(v, w)$ and $d'(u, w) \leq d'(u, v) + d'(v, w)$;
   (c) $u_P \neq v_P$ and $v_P \neq w_P$. Then $d'(u, w) = d(u_P, w_P)$, $d'(v, w) = d(v_P, w_P)$ and since $d(\cdot)$ satisfies the triangle inequality, $d'(u, w) \leq d'(u, v) + d'(v, w)$ holds.

By combining the above results, we conclude that $d'(\cdot, \cdot)$ is a metric distance.  $\square$

## B.4 Proof of Theorem 3

*Proof.* Let MaxMinAlg be a $\beta$-approximation algorithm for $k$-selection under partition matroid constraints for max-min diversification. Consider $\{(P_0, k_0), (B_1, k_1), \ldots, (B_m, k_m)\}$ as the original metric space with individual fairness regions on $P$ constructed by Algorithm 2. Consider an instance of $\alpha$-fair $k$-selection on $P$ and let $c$ be the instance of $k$-selection under partition matroid constraints constructed by Algorithm 3 with input parameters $P, k$, and $\alpha$. We show that the solution returned by MaxMinAlg$(\{(P_0, k_0), (B_1, k_1), \ldots, (B_m, k_m)\})$ is a $(\beta, 3)$-bicriteria approximate solution of the given instance of $\alpha$-fair $k$-selection on $P$.

Let $\text{SOL}_G$ be the solution returned by MaxMinAlg$(\{(P_0, k_0), (B_1, k_1), \ldots, (B_m, k_m)\})$. Let $\text{OPT}_G$ be the optimal solution on $(\{(P_0, k_0), (B_1, k_1), \ldots, (B_m, k_m)\})$ maximizing the diversity for max-min diversification on $P$ under partition matroid constraints. We let $\text{OPT}'_G$ be the optimal solution on $(P', \{(\overline{P}_0, \overline{k}_0), (\overline{B}_1, \overline{k}_1), \ldots, (\overline{B}_m, \overline{k}_m)\}, d')$ from Algorithm 3 maximizing the diversity for max-min diversification on $P'$ under partition matroid constraints. We also let $\text{OPT}_I$ be the optimal solution on $P$ which maximizes the diversity with individual fairness constraints satisfied.

**Fairness Approximation:** Given that $\text{SOL}_G$ is constructed under partition matroid constraints, for each $i \in [m]$, $|B_i \cap \text{SOL}_G| \geq 1$. Hence, by Lemma 2, $\text{SOL}_G$ is a $(3\alpha)$-fair $k$-selection of $P$.

**Diversity Approximation:** We first prove that $\text{div}_{\text{mm}}(\text{OPT}_G, P) = \text{div}_{\text{mm}}(\text{OPT}'_G, P')$. No matter what $\text{OPT}_G$ is, we can always construct $\text{COR}'_G$ in $P'$ whose diversity is equivalent to $\text{OPT}_G$ in $P$. We start with an initially empty set of centers $\text{COR}'_G$. In the first step, for each $B \in \mathcal{B}$, let $c_i$ denote an arbitrary center in $\text{OPT}_G \cap B_i$, and then we add the point $c \in \overline{B}_i$ corresponding to $c_i$ to $\text{COR}'_G$. Next, in the second step, for each $o_0$ in the rest points of $\text{OPT}_G$, we add the point $o \in \overline{P}_0$ corresponding to $o_0$ to $\text{COR}'_G$. It is obvious that $\text{COR}'_G$ has exactly $k$ distinct centers and the pairwise distances between $\text{COR}'_G$ are the same as those between $\text{OPT}_G$. Therefore, $\text{div}_{\text{mm}}(\text{OPT}_G, P) = \text{div}_{\text{mm}}(\text{COR}'_G, P')$. Considering that $\text{div}_{\text{mm}}(\text{OPT}'_G, P') \geq \text{div}_{\text{mm}}(\text{COR}'_G, P')$, we have $\text{div}_{\text{mm}}(\text{OPT}_G, P) \leq \text{div}_{\text{mm}}(\text{OPT}'_G, P')$.

Now we assume that $\text{div}_{\text{mm}}(\text{OPT}_G, P) < \text{div}_{\text{mm}}(\text{OPT}'_G, P')$. If there do not exist two points $u', v'$ in $\text{OPT}'_G$ that are the copies of the same point in $P$, then we can always find the original $k$ points in $P$ (which are corresponding to $\text{OPT}'_G$) whose diversity is greater than current $\text{div}_{\text{mm}}(\text{OPT}_G, P)$ and equals $\text{div}_{\text{mm}}(\text{OPT}'_G, P')$, which is a contradiction. If there exist two points $u', v'$ in $\text{OPT}'_G$ that are the copies of the same point in $P$, then the diversity is $\varepsilon\delta$ given the definition of max-min diversification, which is much smaller than $\text{div}_{\text{mm}}(\text{OPT}_G, P)$, leading to a contradiction. Therefore, $\text{div}_{\text{mm}}(\text{OPT}_G, P) = \text{div}_{\text{mm}}(\text{OPT}'_G, P')$.

Next, we connect the partition matroid constraints on $P'$ with individual fairness constraints on $P$. By the definition of $\alpha$-fairness, each point $v \in P$ must have a center in $\text{OPT}_I$ within distance at most $\alpha \cdot r(v)$. Hence, for each individual fairness region $B \in \mathcal{B}$, $|\text{OPT}_I \cap B| \geq 1$. For each $i \in [m]$, let $c_i$ be the copy of an arbitrary center $c \in \text{OPT}_I \cap B_i$ in the set $\overline{B}_i$. For the remaining points in $\text{OPT}_I$, we pick their corresponding copies in the set $\overline{P}_0$. Let $\text{OPT}'_C$ denote the constructed solution for the instance $P'$. Since $\text{OPT}'_C$ picks exactly one point from each set $\overline{B}_i$, for $i \in [m]$, and exactly $k-m$ points from $\overline{P}_0$, $\text{OPT}'_C$ is a feasible solution for max-min diversification under partition matroid constraints on instance $(P', \{(\overline{P}_0, \overline{k}_0), (\overline{B}_1, \overline{k}_1), \ldots, (\overline{B}_m, \overline{k}_m)\}, d')$. Since the pairwise distances between $\text{OPT}_I$ are the same as those between $\text{OPT}'_C$, we can have $\text{div}_{\text{mm}}(\text{OPT}_I, P) = \text{div}_{\text{mm}}(\text{OPT}'_C, P')$. Considering that $\text{OPT}'_G$ is the optimal solution on instance $(P', \{(\overline{P}_0, \overline{k}_0), (\overline{B}_1, \overline{k}_1), \ldots, (\overline{B}_m, \overline{k}_m)\}, d')$, we can have $\text{div}_{\text{mm}}(\text{OPT}'_G, P') \geq \text{div}_{\text{mm}}(\text{OPT}'_C, P')$. Hence, we have

$$\text{div}_{\text{mm}}(\text{OPT}_I, P) \leq \text{div}_{\text{mm}}(\text{OPT}'_G, P') = \text{div}_{\text{mm}}(\text{OPT}_G, P') \leq \beta \cdot \text{div}_{\text{mm}}(\text{SOL}_G, P).$$

Thus, the diversity of max-min diversification under individual fairness constraints of $P$ using $\text{SOL}_G$ is within a $\beta$ factor of the diversity of any optimal $\alpha$-fair $k$-selection of $P$. $\square$

## B.5 Proof of Theorem 4

*Proof.* We use the same notations as those in Proof of Theorem 3. The only difference is that MaxMinAlg is changed into MaxSumAlg, referring to a $\beta$-approximation algorithm for $k$-selection under partition matroid constraints for max-sum diversification. We show that the solution returned by MaxSumAlg$(\{(P_0, k_0), (B_1, k_1), \ldots, (B_m, k_m)\})$ is a $(\beta(4 + \varepsilon), 3)$-bicriteria approximate solution of the given instance of $\alpha$-fair $k$-selection on $P$.

**Diversity Approximation:** We first prove that $(4 + \varepsilon) \cdot \mathsf{div}_{\mathrm{ms}}(\mathrm{OPT}_G, P) \geq \mathsf{div}_{\mathrm{ms}}(\mathrm{OPT}'_G, P')$. We can get $\mathsf{div}_{\mathrm{ms}}(\mathrm{OPT}_G, P) \leq \mathsf{div}_{\mathrm{ms}}(\mathrm{OPT}'_G, P')$ through the same process shown in the same part in Proof of Theorem 3. Let us focus on the case when $\mathsf{div}_{\mathrm{ms}}(\mathrm{OPT}_G, P) < \mathsf{div}_{\mathrm{ms}}(\mathrm{OPT}'_G, P')$ occurs. In this case, there are two or more points in $\mathrm{OPT}'_G$ being copies of the same point in $\mathrm{OPT}_G$. The extreme situation is that each point $p'_1$ in $\mathrm{OPT}'_G$ can always find another point $p'_2$ in $\mathrm{OPT}'_G$, and their original point in $P$ is the same point. Therefore, in general situations, there are at most $\lfloor \frac{k}{2} \rfloor$ points in $P$ whose two copies in $P'$ are both selected.

In order to obtain the diversity of the extreme situation, we now consider the basic case where there are only four points in $P'$ that are selected, and there are only two points in $P$, which are the original points of the four points in $P'$. W.l.o.g., let $\mathrm{EXP}'_4$ be the four points in $P'$ containing $a'_1, a'_2, a'_3, a'_4$ and $\mathrm{EXP}_2$ be the two distinct original points in $P$ containing $a_1, a_3$, where $a'_1, a'_2$ are the copies of $a_1$, and $a'_3, a'_4$ are the copies of $a_3$. From the definition, we can have $d'(a'_1, a'_2) = d'(a'_3, a'_4) = \varepsilon\delta$, $d'(a'_1, a'_3) = d'(a'_1, a'_4) = d'(a'_2, a'_3) = d'(a'_2, a'_4) = d(a_1, a_3)$. It is obvious that $\mathsf{div}_{\mathrm{ms}}(\mathrm{EXP}_2, P) = d(a_1, a_3)$, while we have $\mathsf{div}_{\mathrm{ms}}(\mathrm{EXP}'_4, P') = d'(a'_1, a'_2) + d'(a'_3, a'_4) + d'(a'_1, a'_3) + d'(a'_1, a'_4) + d'(a'_2, a'_3) + d'(a'_2, a'_4) = 4 \cdot \mathsf{div}_{\mathrm{ms}}(\mathrm{EXP}_2, P) + 2\varepsilon\delta$.

Now consider the situation for $\mathrm{OPT}'_G$ is not extreme, i.e., there exists at least one point $p'_1$ in $\mathrm{OPT}'_G$ that is unable to find another point $p'_2$ in $\mathrm{OPT}'_G$ satisfying $d'(p'_1, p'_2) = \varepsilon\delta$. There are two types of basic situations. W.l.o.g., in the first situation, we let $\mathrm{EXP}'_3$ be the 3 points in $P'$ containing $a'_1, a'_2, a'_3$ and $\mathrm{EXP}_2$ be the 2 distinct original points in $P$ containing $a_1, a_2$, where $a'_2, a'_3$ are the copies of $a_2$, and $a'_1$ is the only copy of $a_1$. From the definition, we can have $d'(a'_2, a'_3) = \varepsilon\delta$, $d'(a'_1, a'_2) = d'(a'_1, a'_3) = d(a_1, a_2)$. We can have $\mathsf{div}_{\mathrm{ms}}(\mathrm{EXP}'_3, P') = d'(a'_1, a'_2) + d'(a'_1, a'_3) + d'(a'_2, a'_3) = 2 \cdot \mathsf{div}_{\mathrm{ms}}(\mathrm{EXP}_2, P) + \varepsilon\delta \leq 4 \cdot \mathsf{div}_{\mathrm{ms}}(\mathrm{EXP}_2, P) + 2\varepsilon\delta$. In the second situation, we let $\mathrm{EXP}'_2$ be the 2 points in $P'$ containing $a'_1, a'_2$ and $\mathrm{EXP}_2$ be the 2 original points in $P$ containing $a_1, a_2$, where $a'_1$ is the only copy of $a_1$, and $a'_2$ is the only copy of $a_2$ as well. We have $d'(a'_1, a'_2) = d(a_1, a_2)$ and $\mathsf{div}_{\mathrm{ms}}(\mathrm{EXP}'_2, P') = \mathsf{div}_{\mathrm{ms}}(\mathrm{EXP}_2, P) \leq 4 \cdot \mathsf{div}_{\mathrm{ms}}(\mathrm{EXP}_2, P) + 2\varepsilon\delta$.

For a general $\mathrm{OPT}'_G$ in $P'$, $\mathsf{div}_{\mathrm{ms}}(\mathrm{OPT}'_G)$ is a combination of the three basic cases mentioned above. We let ORI be the distinct original points in $P$ which are corresponding to $\mathrm{OPT}'_G$ in $P'$. We can have $4 \cdot \mathsf{div}_{\mathrm{ms}}(\mathrm{ORI}, P) + \lfloor \frac{k}{2} \rfloor \varepsilon\delta \geq \mathsf{div}_{\mathrm{ms}}(\mathrm{OPT}'_G, P')$. Given that there at least $\lfloor \frac{k}{2} \rfloor$ points in ORI, we have $\mathsf{div}_{\mathrm{ms}}(\mathrm{ORI}, P) \geq \lfloor \frac{k}{2} \rfloor \delta$, and we can further have $(4 + \varepsilon) \cdot \mathsf{div}_{\mathrm{ms}}(\mathrm{ORI}, P) \geq \mathsf{div}_{\mathrm{ms}}(\mathrm{OPT}'_G, P')$. Next, we assume $|\mathrm{ORI}| = k_s \geq \lfloor \frac{k}{2} \rfloor$, and we let $\mathrm{OPT}_G^{k_s}$ be the optimal solution for $k_s$-point max-sum diversification under partition matroid constraints in $P$. We can have $\mathsf{div}_{\mathrm{ms}}(\mathrm{OPT}_G, P) \geq \mathsf{div}_{\mathrm{ms}}(\mathrm{OPT}_G^{k_s}, P)$, since the addition of any other point to $\mathrm{OPT}_G^{k_s}$ would increase the diversity. Given that $\mathsf{div}_{\mathrm{ms}}(\mathrm{OPT}_G^{k_s}, P) \geq \mathsf{div}_{\mathrm{ms}}(\mathrm{ORI}, P)$, we can have $\mathsf{div}_{\mathrm{ms}}(\mathrm{OPT}_G, P) \geq \mathsf{div}_{\mathrm{ms}}(\mathrm{ORI}, P)$. Therefore, we can come to the conclusion that $(4 + \varepsilon) \cdot \mathsf{div}_{\mathrm{ms}}(\mathrm{OPT}_G, P) \geq \mathsf{div}_{\mathrm{ms}}(\mathrm{OPT}'_G, P')$.

Through the same process as that in proof of Theorem 3, we have

$$\mathsf{div}_{\mathrm{ms}}(\mathrm{OPT}_I, P) \leq \mathsf{div}_{\mathrm{ms}}(\mathrm{OPT}'_G, P') \leq (4 + \varepsilon) \cdot \mathsf{div}_{\mathrm{ms}}(\mathrm{OPT}_G, P) \leq \beta(4 + \varepsilon) \cdot \mathsf{div}_{\mathrm{ms}}(\mathrm{SOL}_G, P).$$

Thus, the diversity of max-sum diversification under individual fairness constraints of $P$ using $\mathrm{SOL}_G$ is within a $\beta(4 + \varepsilon)$ factor of the diversity of any optimal $\alpha$-fair $k$-selection of $P$. $\qquad\square$

## B.6 Proof of Theorem 5

*Proof.* We use the same notations as those in the proof of Theorem 3. The only difference is that MAXMINALG is changed into SUMMINALG, referring to a $\beta$-approximation algorithm for $k$-selection under partition matroid constraints for sum-min diversification. We show that the solution returned by SUMMINALG($\{(P_0, k_0), (B_1, k_1), \ldots, (B_m, k_m)\}$) is a $(\beta(4 + \varepsilon), 3)$-bicriteria approximate solution of the given instance of $\alpha$-fair $k$-selection on $P$.

**Diversity Approximation:** We first prove that $(4 + \varepsilon) \cdot \mathsf{div}_{\mathrm{sm}}(\mathrm{OPT}_G, P) \geq \mathsf{div}_{\mathrm{sm}}(\mathrm{OPT}'_G, P')$. We can get $\mathsf{div}_{\mathrm{sm}}(\mathrm{OPT}_G, P) \leq \mathsf{div}_{\mathrm{sm}}(\mathrm{OPT}'_G, P')$ through the same process shown in the same part in Proof of Theorem 3.

Let us focus on the case when $\mathsf{div}_{\mathrm{sm}}(\mathrm{OPT}_G, P) < \mathsf{div}_{\mathrm{sm}}(\mathrm{OPT}'_G, P')$ occurs. In this case, there are two or more points in $\mathrm{OPT}'_G$ being copies of the same point in $\mathrm{OPT}_G$. We assume that there are $z$ distinct original points in $P$ corresponding to $\mathrm{OPT}'_G$ in $P'$, $z \leq k$. Now we focus on two points $a'_1, a'_2$ in $\mathrm{OPT}'_G$ satisfying $d'(a'_1, a'_2) = \varepsilon\delta$ if they exist. It can be inferred that these kinds of points contribute

$2(k-z)\varepsilon\delta$ to the overall diversity, since $d'(a_1', \mathrm{OPT}_G' \backslash a_1') = d'(a_2', \mathrm{OPT}_G' \backslash a_2') = d'(a_1', a_2') = \varepsilon\delta$ and there are $2(k-z)$ points from $\mathrm{OPT}_G'$ involved in this calculation.

Let us define $D = \mathsf{div}_{\mathrm{sm}}(\mathrm{OPT}_G', P') - 2(k-z)\varepsilon\delta$. We notice that $D$ does not equal the diversity of the original $z$ points in $P$. This is because some points in the original $z$ points that are copied to $\mathrm{OPT}_G'$ have contributed $\varepsilon\delta$ to $\mathsf{div}_{\mathrm{sm}}(\mathrm{OPT}_G', P')$, so they cannot make any other contribution to $D$. Let $\mathrm{OPT}_G^z$ be the optimal solution for $z$-point sum-min diversification under partition matroid constraints in $P$. From what we have discussed above, we can get $D \leq \mathsf{div}_{\mathrm{sm}}(\mathrm{OPT}_G^z, P)$.

Now we are going to introduce a lemma [6] so as to compare $\mathsf{div}_{\mathrm{sm}}(\mathrm{OPT}_G^z, P)$ and $\mathsf{div}_{\mathrm{sm}}(\mathrm{OPT}_G, P)$.

**Lemma 4.** *Let $(P, d)$ be a metric space, and $n = |P|$. Suppose $1 < k < n/3$ is the target number of elements. Let $S'$ be any subset of $V$ of size $\leq k$. Then we can efficiently find an $S \subseteq V$ of size $= k$, such that $\mathsf{div}_{sm}(S, P) \geq \frac{1}{4}\mathsf{div}_{sm}(S', P)$. (See the proof in Appendix B.7.)*

Based on Lemma 4, for $\mathrm{OPT}_G^z$, we can always find a $\mathrm{TEMP} \subseteq P$ of size $= k$, such that $4 \cdot \mathsf{div}_{\mathrm{sm}}(\mathrm{TEMP}, P) \geq \mathsf{div}_{\mathrm{sm}}(\mathrm{OPT}_G^z, P)$. Obviously, $\mathsf{div}_{\mathrm{sm}}(\mathrm{TEMP}, P) \leq \mathsf{div}_{\mathrm{sm}}(\mathrm{OPT}_G, P)$, since $\mathrm{OPT}_G$ is the optimal solution. We can now have

$$\mathsf{div}_{\mathrm{sm}}(\mathrm{OPT}_G^z, P) \leq 4 \cdot \mathsf{div}_{\mathrm{sm}}(\mathrm{OPT}_G, P)$$

if $1 < k < n/3$. Now we have

$$\begin{aligned}
\mathsf{div}_{\mathrm{sm}}(\mathrm{OPT}_G', P') &= D + 2(k-z)\varepsilon\delta \\
&\leq \mathsf{div}_{\mathrm{sm}}(\mathrm{OPT}_G^z, P) + 2(k-z)\varepsilon\delta \\
&\leq 4 \cdot \mathsf{div}_{\mathrm{sm}}(\mathrm{OPT}_G, P) + 2(k-z)\varepsilon\delta.
\end{aligned}$$

As mentioned in the proof of Theorem 4, $k - z \geq \lfloor \frac{k}{2} \rfloor$. Therefore, we have $2(k-z)\varepsilon\delta \leq 2\lfloor \frac{k}{2} \rfloor \varepsilon\delta \leq k\varepsilon\delta \leq \varepsilon \cdot \mathsf{div}_{\mathrm{sm}}(\mathrm{OPT}_G, P)$. Therefore, we have $\mathsf{div}_{\mathrm{sm}}(\mathrm{OPT}_G', P') \leq (4 + \varepsilon) \cdot \mathsf{div}_{\mathrm{sm}}(\mathrm{OPT}_G, P)$.

Hence, we finally come to the conclusion that

$$\begin{aligned}
\mathsf{div}_{\mathrm{sm}}(\mathrm{OPT}_I, P) &\leq \mathsf{div}_{\mathrm{sm}}(\mathrm{OPT}_G', P') \\
&\leq (4 + \varepsilon) \cdot \mathsf{div}_{\mathrm{sm}}(\mathrm{OPT}_G, P) \\
&\leq \beta(4 + \varepsilon) \cdot \mathsf{div}_{\mathrm{sm}}(\mathrm{SOL}_G, P).
\end{aligned}$$

Thus, the diversity of sum-min diversification under individual fairness constraints of $P$ using $\mathrm{SOL}_G$ is within a $\beta(4 + \varepsilon)$ factor of the diversity of any optimal $\alpha$-fair $k$-selection of $P$. $\qquad\square$

### B.7 Proof of Lemma 4

*Proof.* Let us suppose $1 < k < n/3$, and let $S'$ be a set of size $r < k$. We may assume that $r \geq 2$, and so we can suppose that $S' = \{u_1, u_2, \ldots, u_r\}$, for $1 < r < k$.

Let us partition $P$ into $P_1, P_2, \ldots, P_r$, where $P_i$ is the set of vertices whose closest neighbor in $S'$ is $u_i$. Without loss of generality, assume that $|P_1| \geq |P_2| \geq \cdots \geq |P_r|$. Also, let $d_i$ denote the minimum distance from $u_i$ to the rest of $S'$. We consider two cases:

*Case 1 ($|P_1| \geq k$):* First, consider the set of points $W_1 := \{u_2, u_3, \ldots, u_r\} \cup T$, where $T$ is an arbitrary set of $k - r + 1$ points in $P_1$. We claim that the diversity of $W_1$ is at least $(d_2 + d_3 + \cdots + d_r)/2$. This is because for any $i > 1$, every point in $P_1$ is at a distance at least $d_i/2$ from $u_i$ (to see this, consider some $v \in P_i$; we know that $d(v, u_1) \leq d(v, u_i)$, by the definition of $P_i$; thus, if $d(v, u_i) < d_i/2$, we must have $d(v, u_1) < d_i$, a contradiction). Second, let $v$ be the $u_i$ that is furthest from $u_1$. Clearly, we have $d(u_1, v) \geq d_1$. Now consider the set of points $W_2 := \{v\} \cup T$, where $T$ is any set of $k - 1$ vertices in $P_1$. From the same argument as above, the diversity of $W_2$ is at least $d_1/2$. Now, one of the sets above must have diversity $\geq (d_1 + d_2 + \cdots + d_r)/4 \geq \mathsf{div}_{\mathrm{sm}}(S', P)/4$. This completes the argument in this case.

*Case 2 ($|P_1| < k < n/3$):* In other words, all the sets $V_i$ have size $< k$. Now, let $s$ be the smallest index for which $|P_1 \cup \cdots \cup P_s| \geq k$. Since all the $|P_i|$ are smaller than $k < n/3$, we certainly have $s < r$. Furthermore, we must have $|P_{s+1} \cup \cdots \cup P_r| \geq k$. Now, define $W_1 := \{u_1, u_2, \ldots, u_s\} \cup T$, where $T$ is an arbitrary set of $k - s$ elements from $P_{s+1} \cup \cdots \cup P_r$ and $W_2 := \{u_{s+1}, \ldots, u_r\} \cup T'$, where $T'$ is an arbitrary set of $k - s$ elements from $P_1 \cup \cdots \cup P_s$. By the above argument, the diversity of $W_1$ is at least $(d_1 + d_2 + \cdots + d_s)/2$, and that of $W_2$ is at least $(d_{s+1} + \cdots + d_r)/2$. As before, one of these quantities is at least $\mathsf{div}_{\mathrm{sm}}(S', P)/4$. $\qquad\square$

# C  Discussion on Error Parameter in Theorems 6–8

## C.1  Error Parameter in Theorem 6

The algorithm used in Theorem 6 is FMMD-S from [46], which is a $(5 + \varepsilon)$-approximation algorithm for max-min diversification under group fairness constraints running in $O(mkn + m^k \log \frac{1}{\varepsilon})$ time. Therefore, $\varepsilon$ in Theorem 6 directly inherits the error parameter in FMMD-S.

## C.2  Error Parameter in Theorem 7

In the proof of Theorem 4, $\varepsilon$ is an accuracy parameter in Algorithm 3. Therefore, $\varepsilon$ in Theorem 7 is the accuracy parameter in Algorithm 3. In Section 5, $\varepsilon$ is a small positive constant used to control the threshold for accepting local improvements. Thus, a smaller $\varepsilon$ can potentially lead to better results while incurring higher computational overhead.

## C.3  Error Parameter in Theorem 8

In the proof of Theorem 5, $\varepsilon$ is an accuracy parameter in Algorithm 3. The algorithm used in Theorem 8 is from [6] with no other parameter involved in its approximation factor. Therefore, $\varepsilon$ in Theorem 8 is just the accuracy parameter in Algorithm 3.

# D  Extended Theoretical Analysis for Max-Min Diversification

In Theorem 6, we show that the time complexity of our algorithm is exponential w.r.t. $k$, as it uses FMMD-S [46] as a subroutine. This arises because max-min diversification with individual fairness is at least as hard as its partition matroid-constrained version (Theorem 3). To date, however, no algorithm can simultaneously satisfy the following four requirements: (i) the solution exactly meets the partition matroid constraint; (ii) the time complexity is polynomial; (iii) the solution provides a constant-factor approximation guarantee for max-min diversification; and (iv) the algorithm works in general metric spaces.

To provide different trade-offs among these requirements, we show that our algorithm can work with different algorithms for max-min diversification with group fairness constraints. The main results are summarized as follows:

1. **FMMD-S [46]:** According to the concept of group fairness defined in [46], after executing Algorithm 2, one can directly apply FMMD-S by restricting the number of points selected in each individual fairness region to lie between $1$ and $k - m + 1$. The solution achieves a 5-approximation while strictly satisfying group fairness constraints, as indicated in Theorem 6. However, FMMD-S relies on solving an ILP for solution computation and thus has an exponential time complexity w.r.t. $k$.

2. **Fair-Greedy-Flow [3]:** According to the concept of group fairness defined in [3], after executing Algorithm 2, one can directly apply Fair-Greedy-Flow by pre-specifying for each individual fairness region a constant $k_i \geq 1$ such that the sum of all $k_i$'s is equal to $k$. This algorithm runs in $O(nkm^3)$ time but only provides an $O(m)$-approximation, where $m$ is the number of individual fairness regions.

3. **MFD [25]:** According to the definition of group fairness in [25], after executing Algorithm 2, one can directly apply MFD by pre-specifying for each individual fairness region a constant $k_i \geq 1$ such that the sum of all $k_i$'s is equal to $k$. This algorithm has a near-linear time complexity of $O(nk)$ and achieves a constant approximation. However, the algorithm is randomized and its solutions may not always satisfy the fairness constraints. In addition, MFD is specific to Euclidean space.

Each of these methods does not meet at least one of the four requirements. Considering the empirical performance of our algorithm, we chose the FMMD-S algorithm [46] in our implementation.

