# OpenReview forum: "Individually Fair Diversity Maximization"
_NeurIPS.cc/2025/Conference — NeurIPS 2025 poster_

### Official Review · Reviewer_15en · 2025-06-28

**Clarity:** 3
**Significance:** 2
**Originality:** 3
**Rating:** 4
**Confidence:** 4

**Summary:**

The paper discusses the problem of finding a diverse set of k points from a given point set in a metric space. There are three ways of quantifying diversity:
1. max-min diversification: maximize the minimum distance between any pair of selected points,
2. max-sum diversification: maximize the sum of pairwise distances between selected points, and
3. sum-min diversification: maximize the sum of the minimum distances from each selected point to its nearest neighbor among the other selected points.

These problems are well-studied in the literature, and an approximation lower bound of 2 is known for them. Approximation algorithms closely matching the lower bound are also known. This paper explores these problems with an additional constraint that every point in the point set has at least one of its (n/k)-nearest neighbors as its “representative” among the chosen k points. This property is called the individually fair property.

The paper gives approximation algorithms for the three diversity problem with the added individually fair constraint. The nature of the approximation algorithm is a bicriteria (a, b) approximation, where a is the approximation guarantee for the diversity objective, and b is the factor by which the individually fair radius is exceeded.

The algorithm uses the following reduction-based idea:
- First, the set of points is partitioned into (approximate) individually fair regions.
- Then, a known approximation algorithm for the diversity problem under the partition matroid is used on the point set and the partition to obtain the bicriteria approximation algorithm.

Experiments are performed and observations are made about the algorithms with and without the individually fair constraint.

**Questions:**

Comments:
- (lines 61 - 75): Bicriteria approximation has not been defined in the paper yet. It is only defined later. So, it is difficult to read. Also, what purpose does \alpha serve in these statements?
- Other questions are mentioned in the summary and strengths and weaknesses.

**Ethical Concerns:**

["NO or VERY MINOR ethics concerns only"]

**Final Justification:**

I remain positive about the paper after the rebuttal. The authors agree to discuss some of the points I raised in the final version. I will maintain my original score.

**Limitations:**

Yes. The paper presents theoretical results that do not have any potential negative societal impact.

**Paper Formatting Concerns:**

There are no serious paper formatting concerns.

**Quality:**

3

**Strengths And Weaknesses:**

Strengths:
- Theoretical results about two opposing objectives (diversity and individual fairness) are good to know. The problem formulation makes sense and may be useful.


Weaknesses:
- Since the major part of this work is theoretical, the lack of approximation lower bound statements to support some of the approximation upper bounds stands out and can be seen as a weakness.
- Discussions related to checking whether there is a feasible solution for the individually fair problem seem to be missing. It'll be good to have this discussion.
- The experimental results are designed to observe properties of the problem and the suggested algorithm. Some experiments that can motivate the problem formulation can help to make the paper more interesting.

---

> ### Author Rebuttal · Authors · 2025-07-30
>
> We are truly grateful for your insightful feedback and the time you spent evaluating our work. Below, we provide detailed responses to each of your comments.
>
> > Q1: Since the major part of this work is theoretical, the lack of approximation lower bound statements to support some of the approximation upper bounds stands out and can be seen as a weakness.
>
> A1: Thank you for pointing this out. We have addressed this concern in our response to Reviewer NtvU’s Q2, where we prove that for all three diversity maximization objectives under individual fairness constraints, the approximation lower bound cannot be better than 2. We will include the corresponding lower-bound statements and proofs in the revised manuscript to explicitly demonstrate the hardness of our problem.
>
>
>
> > Q2: Discussions related to checking whether there is a feasible solution for the individually fair problem seem to be missing. It'll be good to have this discussion.
>
> A2: We appreciate the reviewer’s suggestion. In the current version, we briefly mention the NP-hardness of deciding whether there exists a solution that strictly satisfies individual fairness after Definition 2 (Lines 152–155). However, we agree that this important issue deserves more explicit discussion. As also pointed out by Reviewer nRw7, it is known that when $\alpha = 2$, a feasible solution always exists, and this bound is tight (see Jung et al. 2019). We will revise the paragraph following Definition 2 to clearly explain the computational hardness of checking feasibility, and to emphasize the significance of the $\alpha = 2$ threshold, which guarantees feasibility.
>
>
>
> > Q3: The experimental results are designed to observe properties of the problem and the suggested algorithm. Some experiments that can motivate the problem formulation can help to make the paper more interesting.
>
> A3: Thank you for your constructive comment. We acknowledge that although our original experiments were designed to demonstrate the effectiveness of our algorithm, their role in motivating the problem formulation was not explicitly conveyed.
>
> We would also like to take this opportunity to reiterate the central motivation of our work: individual fairness in representative selection. That is, our goal is to ensure that each unselected point is adequately represented by the selected set, preventing certain regions of the data from being disproportionately neglected.
>
> To further support the motivation, we have added a new set of experiments to emphasize how fairness constraints influence the solution quality in alignment with the motivation behind our problem definition.
>
> Specifically, we compare the solutions obtained under unconstrained diversity maximization with those obtained under individual fairness constraints, across three canonical diversity objectives. For each case, we evaluate the solutions using three complementary metrics:
>
> 1. **Fairness performance** $⁡\alpha_{\max}$: fairness performance, which has been defined in the paper (Section: *Measures* in *Experiment*, Lines 339–340).
> 2. **Max Distance**: the largest distance between any unselected point and its nearest selected point.
> 3. **Average Distance**: the average distance from each unselected point to its nearest selected point.
>
> Smaller values of these metrics indicate better individual-level representativeness of the selected set, thereby aligning with the goals of individual fairness.
>
> The table below reports partial results from our experiments on the CelebA dataset, with selection sizes $k = 5, 10, 20$. For each diversity objective (Max-Min, Max-Sum, Sum-Min), we compare the algorithms for the unconstrained and fairness-constrained problems using the three metrics mentioned above. These results illustrate how fairness constraints improve individual coverage, which directly supports the motivation for our fairness-aware formulation. All algorithms and other settings used here are the same as those employed in the experimental section of the paper.
>
> | k    | Metric           | Max-Min (Unconstrained) | Max-Min (Fairness) | Max-Sum (Unconstrained) | Max-Sum (Fairness) | Sum-Min (Unconstrained) | Sum-Min (Fairness) |
> | ---- | ---------------- | ----------------------- | ------------------ | ----------------------- | ------------------ | ----------------------- | ------------------ |
> | 5    | Actual α         | 1.18                    | **1.05**           | 1.30                    | **1.15**           | 1.17                    | **1.10**           |
> |      | Max Distance     | 2019.70                 | **1987.75**        | 2100.38                 | **2075.56**        | **2029.92**             | 2039.88            |
> |      | Average Distance | 1306.51                 | **1193.54**        | 1406.64                 | **1291.87**        | 1297.98                 | **1235.41**        |
> | 10   | Actual α         | 1.27                    | **1.10**           | 2.16                    | **1.24**           | 1.26                    | **1.13**           |
> |      | Max Distance     | 1951.72                 | **1926.83**        | 2359.95                 | **2023.46**        | 1959.41                 | **1939.41**        |
> |      | Average Distance | 1303.03                 | **1173.17**        | 1941.29                 | **1270.74**        | 1293.79                 | **1211.71**        |
> | 20   | Actual α         | 1.35                    | **1.18**           | 2.121                   | **1.32**           | 1.35                    | **1.25**           |
> |      | Max Distance     | 1822.25                 | **1771.05**        | 2167.25                 | **1899.74**        | **1833.73**             | 1849.33            |
> |      | Average Distance | 1296.87                 | **1156.07**        | 1766.35                 | **1264.50**        | 1295.00                 | **1220.42**        |
>
> The table above demonstrates that across all diversity objectives and selection sizes, the fairness-constrained algorithms consistently outperform or match their unconstrained counterparts in terms of both fairness and representation quality. These consistent improvements support our central argument that enforcing fairness constraints enhances individual coverage without sacrificing overall selection quality.
>
> Due to the space and time constraints of the rebuttal, we only report partial results here. In the revised version of the paper, we will provide the full experimental settings, including results across all datasets mentioned in the manuscript and for a wider range of $k$ values, along the three metrics above. This extended analysis will further highlight our motivation and confirm that our algorithms indeed improve individual fairness compared to their unconstrained counterparts.
>
>
>
> > Q4: Bicriteria approximation has not been defined in the paper yet. It is only defined later. So, it is difficult to read. Also, what purpose does \alpha serve in these statements?
>
> A4: Thank you for highlighting this point. The parameter $\alpha$ controls the level of individual fairness — specifically, it allows for relaxing the strict fairness constraint in order to achieve approximate feasibility. To improve readability, we will add intuitive and informal explanations of both $\alpha$ and bicriteria approximation in **Introduction**, so that readers can better understand their role before encountering the formal definitions.
>
>
>
> Thank you again for your constructive comments!

---

> > ### Comment · Reviewer_15en · 2025-08-04
> >
> > I would like to thank the authors for the detailed response. I have gone over the rebuttal carefully. The lower bounds pointed out by the authors are for the diversification problem (and hence will also hold for the new problem). I was suggesting lower bounds for the new problem. The approximation guarantees are worse than 5. So, does it mean that the diversification problems, along with the fairness constraint, are harder problems from the point of approximation hardness? So, questions of the form "is (5, 3) the best one can hope for max-min diversification under individual fairness?" This does not get addressed. I think the paper has some nice results. So, I will maintain my positive score.

---

> ### Author Response · Authors · 2025-08-04
> **Thank You for Highlighting the Open Question on Approximation Hardness!**
>
> We sincerely thank you for your thoughtful and encouraging feedback! Your comments on approximation lower bounds for the new problem are highly insightful and have given us valuable inspiration for future research directions.
>
> We acknowledge that obtaining tighter bounds under individual fairness constraints is challenging, as the problem inherently involves a delicate trade-off between diversity and the fairness parameter $\alpha$. Most previous works on group-fair diversity maximization are also limited to deriving a $2$-approximation lower bound based on the unconstrained formulation (Wang et al. (2023) *Max-Min Diversification with Fairness Constraints: Exact and Approximation Algorithms*, Moumoulidou et al. (2020) *Diverse Data Selection under Fairness Constraints*). Exploring whether results such as the $(5,3)$-bicriteria approximation for max-min diversification under individual fairness are optimal is indeed an important open question. We will take this as a key point to further investigate in our future work. If possible, we will add this as a limitation in **Conclusion** section of the revised version to explicitly acknowledge the gap.
>
> Thank you again for raising this insightful point, which has helped us improve both the clarity and completeness of our work.

---

### Official Review · Reviewer_nRw7 · 2025-07-01

**Clarity:** 3
**Significance:** 2
**Originality:** 3
**Rating:** 4
**Confidence:** 3

**Summary:**

The paper studies a combination of diversity maximization and individual fairness in centroid clustering. Diversity maximization requires that the selected points are (in some sense) sufficiently spread out, while individual fairness requires that every point in the clustering instance should have a selected center somewhere in their vicinity. The paper gives a general framework for approximating diversity subject to approximate individual fairness. This framework guarantees bi-criteria approximations to both.

**Questions:**

None currently

**Ethical Concerns:**

["NO or VERY MINOR ethics concerns only"]

**Final Justification:**

Thank you to the authors for the very detailed rebuttal (in fact I think it is too detailed). However, the level of detail still does not entirely convince me that the paper is strong enough for NeurIPS. Just combining two known concepts does not necessarily make for a good/well-motivated paper, and at the moment this paper mostly seems derivate/not strong enough for NeurIPS to me.

**Limitations:**

Yes

**Paper Formatting Concerns:**

/

**Quality:**

3

**Strengths And Weaknesses:**

In general this seems like a solid paper. Both diversity maximization and individual fairness are problems that are somewhat frequently studied, and combining them seems natural. There is a lot of work on bi-criteria optimization for standard clustering objectives (such as k-center or k-means), so why not design bi-criteria optimization algorithms for diversity maximization. In its techniques the paper is very similar to those previous works (which is not really surprising as for instance k-center and Max-Min Diversification are closely related, with the tight $2$-approximations for either essentially being the same algorithm). For instance, the construction of the individual fairness regions here is essentially the same as what is done in ``A Subquadratic Time Approximation Algorithm for Individually
Fair k-Center'' by Ebbens et al. at AISTATS this year (I briefly thought that one might be able to directly apply their algorithm here as well, but this is not as easy, as the algorithm of Ebbens et al. can select strictly less than $k$ points and filling after these points is crucial) and I think a bit more discussion on why previous algorithms fail would have been nice.

In general, while I think that the paper is solid, I am not entirely sure if it is strong enough for NeurIPS. In particular, the paper does not feel very significant to me, and more like another twist on an already mostly established formula. I am willing to be convinced by the other reviewers here, though.

Minor Comments:
- Line 50: "an individual notion of fairness" reads a bit weird to me
- Line 50-51: You should probably note that such a clustering does not always exist
- Line 54: How does this example related to clustering? While this is a "subset-selection" problem it is not really a clustering problem. Also I am not really sure you want your measures of diversity in such a setting.
- Our Contributions: \alpha is undefined here
- Line 109: Technically the group fairness notion studied by paper [10] is the nearly equivalent to individual fairness as studied in this paper, see "Proportional fairness in clustering: A social choice perspective", Kellerhals and Peters, NeurIPS 2024
- "seminal work" might be too high praise
- Line 123: \mapsto should be \to
- Line 125: "to denote the subset of points." what do you mean by "the subset of points"?
- Line 131 onwards: You do not really need P in the definitions, also missing {} around the u(s)
- Line 152: You should briefly mention that alpha = 2 can be achieved and that this is tight.
- I found the statements of Theorems 1, 2,3 to be somewhat unclean, maybe they could be written nicer.
- Line 531: Mathmode around n/k missing

---

> ### Author Rebuttal · Authors · 2025-07-30
>
> Thank you for the positive and constructive comments! Here, we offer comprehensive responses to each of your points.
>
> > Q1:  For instance, the construction of the individual fairness regions here is essentially the same as what is done in ``A Subquadratic Time Approximation Algorithm for Individually Fair k-Center'' by Ebbens et al. at AISTATS this year and I think a bit more discussion on why previous algorithms fail would have been nice.
>
> A1: We appreciate the reviewer’s insightful comment. Indeed, we have carefully considered whether the algorithm proposed by previous work could be directly applied to our setting. However, due to the fundamental differences between our problem and the standard individually fair k-center formulation, we believe such a direct application is not feasible.
>
> The core distinction lies in the objective function. Prior works on fair clustering, including Ebbens et al., focus on minimizing a clustering-based loss, which inherently depends not only on the selected points but also on the structure of the entire dataset. In contrast, diversity maximization objectives—such as max-min, max-sum, or sum-min—depend solely on the selected subset. This creates a fundamental computational divergence between clustering and diversity maximization.
>
> Moreover, diversity objectives exhibit different and often opposite monotonicity properties with respect to the number of selected points. For instance, max-min diversity is typically non-increasing, max-sum diversity is strictly increasing, and sum-min diversity shows no clear monotonic trend. On the other hand, most clustering objectives, including those studied by Ebbens et al., are non-increasing with the number of centers, and are framed as minimization problems. This misalignment in objective structure and monotonicity behavior makes the design and analysis of approximation algorithms for diversity maximization under fairness constraints significantly more challenging.
>
> Therefore, although some high-level conceptual parallels exist, applying previous fair clustering algorithms would require entirely new analytical frameworks and cannot be done in a black-box manner. We hope this clarification highlights the novelty and technical contributions of our approach.
>
> > Q2: In particular, the paper does not feel very significant to me, and more like another twist on an already mostly established formula.
>
> A2: As mentioned in our response to A1, our problem setting is meaningfully distinct from prior works, although we do draw conceptual inspiration from the broader literature on individual fairness.
>
> We would like to emphasize that our contribution is not merely a minor modification of an existing framework. For each diversity objective considered in our paper, we designed approximation algorithms with rigorously proven guarantees, and each proof was carefully tailored to the unique structural properties of the respective objective function. These proofs were developed independently and are not based on prior templates or proof techniques from the literature.
>
> For instance:
>
> - In the Max-Min objective, we leverage the *non-strictly decreasing* property of the diversity function to repeatedly apply proof by contradiction in bounding the approximation factor.
> - In the Max-Sum objective, rather than analyzing sums over arbitrary pairs, we reformulate the analysis in terms of minimal four-point units, which enables a tractable and effective upper bound.
> - In the Sum-Min objective, we address the lack of monotonicity by reducing the analysis to a special case that admits a form of monotonicity proven in our Lemma 4, allowing for a meaningful approximation guarantee.
>
> We believe these contributions—both in problem formulation and in the technical development of distinct analysis tools—demonstrate the novelty and significance of our work.
>
> > Q3: Line 50: "an individual notion of fairness" reads a bit weird to me
>
> A3: The phrase *“an individual notion of fairness”* was used intentionally, following its usage in Vakilian and Yalçıner (2022), *Improved Approximation Algorithms for Individually Fair Clustering*. As our work shares a similar focus on fairness at the individual level we adopted this phrasing to align with established terminology in the literature.
>
> > Q4: Line 50-51: You should probably note that such a clustering does not always exist
>
> A4: In Definition 2 (Lines 150–155) of our paper, we explicitly state that determining whether such a clustering exists is NP-hard. Our intention in the earlier discussion was not to assert the universal existence of such a clustering, but rather to introduce a natural and intuitive formulation of the problem that can help readers grasp the motivation and definition of our setting.
>
> > Q5: Line 54: How does this example related to clustering? While this is a "subset-selection" problem it is not really a clustering problem. Also I am not really sure you want your measures of diversity in such a setting.
>
> A5: Upon reviewing our manuscript, we acknowledge that the description in Line 49-54 was inaccurate. The example provided is indeed a subset-selection problem, not a clustering problem. We will revise the phrasing in this part to refer to it as a diversity maximization problem instead of a clustering problem.
>
> We also recognize that the corporate recruitment example is not the best fit for illustrating diversity maximization under individual fairness, since recruitment is often better modeled as a ranking task. A more appropriate example is the facility location problem, where one must determine a set of facility locations from a large geographical area. For a more detailed justification of this example, we kindly refer the reviewer to our responses to Reviewer uok7’s Q1 and Q2. We will revise the manuscript accordingly and replace the current example with this more suitable one.
>
> > Q6: Our Contributions: \alpha is undefined here
>
> A6: Thank you for pointing this out. The parameter α was indeed not defined when first mentioned in the "Our Contributions" section. In the revised version, we will give intuitive and informal explanations in Introduction, describing α as the fairness parameter that controls the allowed relaxation of individual fairness constraints.
>
> > Q7: Line 109: Technically the group fairness notion studied by paper [10] is the nearly equivalent to individual fairness as studied in this paper
>
> A7: Thank you for the correction. You are absolutely right that [10] aligns more closely with the definition of individual fairness. We will revise the manuscript to relocate [10] alongside the reference you suggested—*Proportional Fairness in Clustering: A Social Choice Perspective* (Kellerhals and Peters, NeurIPS 2024)—in the section where we describe individual fairness more accurately.
>
> > Q8: Line 110: "seminal work" might be too high praise
>
> A8: We described the cited work as “seminal” because it was the first to formally introduce the notion of individual fairness. Similar phrasing can be found in Vakilian and Yalçıner (2022), *Improved Approximation Algorithms for Individually Fair Clustering*, where their introduction refers to the work of Chierichetti et al. (2017) *Fair clustering through fairlets*, which is the first work introducing the concept of fair clustering, in a comparable manner. Nonetheless, if the wording feels overly strong, we are open to softening it in the revision.
>
> > Q9: Line 123: \mapsto should be \to
>
> A9: After reviewing several related works, we agree with this point and we will replace '\mapsto' with '\to'.
>
> > Q10: Line 125: "to denote the subset of points." what do you mean by "the subset of points"? Line 131 onwards: You do not really need P in the definitions, also missing {} around the u(s)
>
> A10: Regarding the missing `{}`, we believe our notation is acceptable and consistent with prior literature. For instance, Bhaskara et al. (2016) in their NIPS paper *Linear Relaxations for Finding Diverse Elements in Metric Spaces* adopt a similar notation in their definition of diversity functions.
>
> As for the use of $P$ in our definitions, we intentionally specify that $S \subseteq P$ and include $P$ when defining the diversity function. This is because in the proofs of Theorems 1, 2, and 3, we construct a new metric space $P'$, and it becomes important to clearly distinguish between the original and modified ground sets. Explicitly stating the ground set $P$ and subset $S$ helps clarify the domain on which diversity is measured and avoids confusion when comparing diversity values across different spaces. Therefore, we prefer to keep the current formulation.
>
> > Q11: Line 152: You should briefly mention that alpha = 2 can be achieved and that this is tight.
>
> A11: We agree that it is helpful to include this information. In Jung et al. (2019), *Service in Your Neighborhood: Fairness in Center Location*, it is shown that an approximation factor of $\alpha = 2$ can be achieved and that this bound is tight. We will add a brief remark to clarify this in the revised version in order to further discuss feasibility.
>
> > Q12: I found the statements of Theorems 1, 2, 3 to be somewhat unclean, maybe they could be written nicer.
>
> A12: Thank you for the suggestion. We understand that the current statements of Theorems 1, 2, and 3 may appear slightly verbose or indirect. We will revise them to improve clarity and conciseness. For example, Theorem 1 can be rewritten as:
>
> *Suppose that there exists a $\beta$-approximation algorithm for $k$-selection under partition matroid constraints for max-min diversification. Then, there exists a $(\beta,3)$-bicriteria approximation for $\alpha$-fair $k$-selection for max-min diversification.*
>
> We will apply similar improvements to Theorems 2 and 3 in the revised version.
>
> Q13: Line 531: Mathmode around n/k missing
>
> A13: We agree this is a missing, and we will replace 'n/k' with '$n/k$'.
>
>
>
> Thank you again for your constructive comments!

---

### Official Review · Reviewer_uok7 · 2025-07-03

**Clarity:** 3
**Significance:** 2
**Originality:** 3
**Rating:** 4
**Confidence:** 4

**Summary:**

The paper focused on the diversity maximization problem with the additional constraint of individual fairness based neighborhood radii as introduced in Jung et al. Three diversity maximization objectives are considered (max-min, max-sum, sum-min) and approximation bicriteria algorithms with guarantees are introduced for them.

**Questions:**

Please see weaknesses above.

**Ethical Concerns:**

["NO or VERY MINOR ethics concerns only"]

**Final Justification:**

The authors have given a better motivation for the problem and will be explicit about the exponential dependence on $k$, this improves the quality of the submission.

**Limitations:**

No real need to discuss potential negative societal impact.

**Paper Formatting Concerns:**

No Formatting Concerns.

**Quality:**

2

**Strengths And Weaknesses:**

## Strengths

-The paper is mostly well-written.

-I did not have a chance to look at the proofs, but the derivations do not seem trivial.

## Weaknesses

I think the major issues are with the motivation for the model/notion of fairness and the parameter $m$ and the runtime in Theorem 4. More details:

1-The motivation of the problem is not clear. Why is this notion of fairness needed? The objectives (1-3) on page 3 are all calculated based on S, but definition 2 on page 4 is based on all of the Points in P not in S? The paper does not give any motivations for the model.

2-lines (52-57), I don’t see how the corporate recruitment example is a good motivation for additionally imposing the n/k constraint? How does it guarantee that every qualified applicant has a chance of being interviewed?

3-There is an issue with Theorem 4. In particular, the run-time is exponential in k which is a big disadvantage. Further, I cannot understand the bound on the value of $m$ , is it not the case that $m \leq k$ . This would make the result significantly weaker. If the run-time can indeed be very slow in terms of k, then I think the paper should've been clear about this. Especially, since the other objectives do not have a similar problem.

4-Is the experimental section comparing against any baselines? I don’t see where the baselines are?


Minor Points:

1-‘Non-private’ line 13, replace by something like unfair instead.

2-In definition 3 on page 4, for point 2. It should be just $\gamma$ not $\gamma \dot \alpha$, correct?

3-What is the GMM algorithm in line 323?

---

> ### Author Rebuttal · Authors · 2025-07-30
>
> We greatly appreciate your thoughtful and constructive comments! Below, we provide detailed responses to each of your points.
>
> > Q1: The motivation of the problem is not clear. Why is this notion of fairness needed? The objectives (1-3) on page 3 are all calculated based on S, but definition 2 on page 4 is based on all of the Points in P not in S? The paper does not give any motivations for the model.
>
> A1: Our motivation focuses on performing diverse data selection, where the goal is to obtain subsets that achieve both strong diversity and individual fairness. Data selection is a common and important task, but identifying such representative subsets is not trivial. Specifically, diversity seeks to maximize dissimilarity among selected items so that the subset captures a wide range of information in the dataset. Fairness, on the other hand, ensures that representation is balanced across certain features.
>
> One application can be found in facility location problems. There, individual fairness requires that every demand point lies within a reasonable distance of some selected facility, ensuring that no individual is left underserved. At the same time, diversity encourages the selected facilities to be well-dispersed across the space, avoiding excessive concentration in one area. This illustrates how fairness and diversity capture complementary goals that must be balanced in representative selection tasks.
>
> It is worth noting that while diversity maximization with group fairness constraints has been studied, individual fairness constraints differ in two critical ways:
>
> 1. **No external group definitions are required.** Fairness regions in our setting are induced directly from data features rather than externally specified demographic attributes.
> 2. **Different perspectives.** Group fairness ensures balance at the group level, but may still neglect individuals within a group. Individual fairness starts from each single data point, ensuring that all individuals are adequately represented by the selected set.
>
> We appreciate this observation, and we will revise the introduction to emphasize these distinctions and make the motivation for our formulation more explicit.
>
> > Q2: Lines (52-57), I don’t see how the corporate recruitment example is a good motivation for additionally imposing the n/k constraint? How does it guarantee that every qualified applicant has a chance of being interviewed?
>
> A2: The corporate recruitment example provided in the manuscript does not fully align with the motivation of individual fairness in diversity maximization, as recruitment is often more naturally modeled as a ranking task rather than a representative selection problem. Therefore, we propose a more suitable example to better illustrate our motivation.
>
> A more suitable example, as discussed in our response to Q1, comes from **facility location**. In this setting, individual fairness requires that every demand point lies within a reasonable distance of some chosen facility, ensuring that no individual is left completely unserved. At the same time, diversity encourages the selected facilities to be spread out rather than concentrated in one area. The $n/k$ constraint reflects this balance: it ensures that representatives are not overly clustered and that coverage is reasonably distributed across the entire population.
>
> For a detailed elaboration of this motivation, we kindly refer the reviewer to our response to Q1. We will revise the manuscript accordingly and replace the current example with this more suitable one.
>
> > Q3: There is an issue with Theorem 4. In particular, the run-time is exponential in *k* which is a big disadvantage. Further, I cannot understand the bound on the value of *m* , is it not the case that *m*<=*k*. This would make the result significantly weaker. If the run-time can indeed be very slow in terms of k, then I think the paper should've been clear about this. Especially, since the other objectives do not have a similar problem.
>
> A3: We agree that the exponential dependence on *k* in the runtime of Theorem 4 is a significant theoretical limitation. As noted in Lemma 1 (Lines 203–204, 522–532), we have formally shown that *m* ≤ *k*, which indeed implies that the runtime complexity is exponential in *k*—a fact that makes the computational cost appear even less desirable.
>
> This arises because max-min diversification with individual fairness is at least as hard as the partition matroid-constrained version (Theorem 1, Line 236-239). To date, however, no algorithm is known that simultaneously (i) satisfies partition matroid constraints exactly, (ii) operates in polynomial time, and (iii) achieves a constant-factor approximation guarantee for max-min diversification in (iv) general metric spaces.
>
> Several representative approaches illustrate the trade-offs:
>
> - **Wang et al. (2023), Max-Min Diversification with Fairness Constraints: Exact and Approximation Algorithms**:
>   According to the definition of group fairness in that paper, after executing Algorithm 2 from our paper, one can directly apply their algorithm **FMMD-S** by restricting the number of points selected in each individual fairness region to lie between $1$ and $k-m+1$. This method achieves an $O(1)$ approximation factor while strictly satisfying group fairness constraints, but relies on solving an ILP formulation, leading to exponential runtime in *k*.
> - **Addanki et al. (2022), Improved Approximation and Scalability for Fair Max-Min Diversification**:
>   According to the definition of group fairness in that paper, after executing Algorithm 2 from our work, one can directly apply their algorithm **Fair-Greedy-Flow** by pre-specifying for each individual fairness region a constant $k_i \geq 1$ such that the sum of all $k_i$ equals $k$. This method runs in polynomial time ($O(nkm^3)$) but only provides an $O(m)$ approximation factor.
> - **Kurkure et al. (2024), Faster Algorithms for Fair Max-Min Diversification in $ℝ^d$**:
>   According to the definition of group fairness in that paper, after executing Algorithm 2 from our work, one can directly apply their algorithm **MFD** by pre-specifying for each individual fairness region a constant $k_i \geq 1$ such that the sum of all $k_i$ equals $k$. This method obtains near-linear runtime $O(nk)$ and constant approximation, but the algorithm is randomized and may not always strictly satisfy fairness constraints. Besides, it is only for Euclidean space.
>
> Each of these alternatives sacrifices some combination of fairness guarantees, approximation quality, runtime, or the type of metric space in which the result holds. Given these trade-offs—and considering the practical performance of our algorithm (See Figure 3)—we chose the Wang et al. algorithm for our implementation.
>
> We acknowledge that the lack of a universally efficient and optimal algorithm for this class of problems is a limitation of our work. We will revise the manuscript in **Conclusion** (Lines 376–379) to clarify this issue and highlight the complexity aspect as an important direction for improvement.
>
> > Q4: Is the experimental section comparing against any baselines? I don’t see where the baselines are?
>
> A4: Our experiments do include baselines. Specifically, in our experimental design, to evaluate the approximation quality of our algorithm with respect to diversity, we measure the ratio between the diversity achieved by our algorithm and that achieved by the unconstrained algorithms. Thus, the unconstrained algorithms serve as baselines to illustrate the *price of fairness*, and our results confirm the effectiveness of our approach.
>
> That said, we acknowledge that we could make the use of baselines more explicit by directly reporting additional metrics obtained from the unconstrained algorithms. As we mention in our response to Reviewer 15en’s Q3, we have added new experiments and results to highlight these comparisons, and we will incorporate these experiments into the revised version of the paper.  We believe this addition will strengthen the persuasiveness of our experimental evaluation.
>
> > Q5: ‘Non-private’ line 13, replace by something like unfair instead.
>
> A5: We agree with this point and we will replace 'Non-private' with 'unfair'.
>
> > Q6: In definition 3 on page 4, for point 2. It should be just *γ* not *γα*, correct?
>
> A6: Upon close inspection, we confirm that our use of *γα* in Definition 3 is intentional and correct. This is because we allow the parameter *α* to be greater than 1, even though *α* = 1 represents the most stringent case. This form of the definition is also consistent with prior work, including Mahabadi and Vakilian (2020)'s *Individual Fairness for k-Clustering*, Vakilian and Yalçıner (2022)'s *Improved Approximation Algorithms for Individually Fair Clustering* and Ebbens et al. (2025)'s *A Subquadratic Time Approximation Algorithm for Individually Fair k-Center*.
>
> > Q7: What is the GMM algorithm in line 323?
>
> A7: The GMM algorithm originates from the work of Gonzalez (1984), *Clustering to Minimize the Maximum Intercluster Distance*, and was later shown in Ravi et al. (1992), *Heuristic and special case algorithms for dispersion problems*, to be a 2-approximation algorithm for Max-Min diversity maximization in metric spaces. We agree that the first occurrence of the term “GMM” lacked an explicit description of the algorithm and its references. We will add the appropriate citation at Line 323 and include a description and pseudocode of the algorithm in the appendix.
>
>
>
> Thank you again for your constructive comments!

---

> > ### Comment · Reviewer_uok7 · 2025-08-04
> >
> > I thank the authors for the rebuttal. Specifically, clarifying the issue with $m$. Also, the motivation/application for the problem.
> >
> > The authors have given an application based on facility location. While it is plausible that one would like to spread the facilities to avoid concentration in areas. All facility location formulations I'm familiar with involve minimizing travel distance between points and facilities. Are application of diversity maximization for facility location considered standard/used or motivated commonly?

---

> ### Author Response · Authors · 2025-08-04
> **Applications of Diversity Maximization in Facility Location are Well-Established.**
>
> Thank you very much for your thoughtful feedback and for raising this important question. Specifically, you asked:
>
> > Are applications of diversity maximization for facility location considered standard/used or motivated commonly?
>
> Our answer is **yes** — the use of diversity maximization in facility location is standard, widely used, and strongly motivated. This has been surveyed in the literature, e.g., Parreño et al. (2020). *Measuring diversity. A review and an empirical analysis*, which discusses many works applying the diversity functions studied in our paper to facility location problems. Here, we highlight several representative examples that are directly relevant to the motivation and methods of our work:
>
> - **Erkut, E. (1990). The discrete p-dispersion problem.**
>   This paper introduced the discrete *p*-dispersion (*max-min diversification*) problem, with the motivation that in applications such as fast-food franchises, decision-makers would not want two franchises located too close to each other, since they would share the same customer base. Instead, spreading them out increases total sales.
> - **Kuby, M. J. (1987). Programming models for facility dispersion: The p‐dispersion and maxisum dispersion problems.**
>   As shown in the title, this work considered the facility location problem using both *p*-dispersion (*max-min diversification*) and maxisum dispersion (*max-sum diversification*).
> - **Chandra, B., & Halldórsson, M. M. (2001). Approximation algorithms for dispersion problems.**
>   This work used facility location as a central motivating application, and analyzed dispersion functions such as Remote-edge, Remote-clique, and Remote-pseudoforest, which correspond to the *max-min*, *max-sum*, and *sum-min diversification* objectives in our work.
> - **Ravi, S. S., Rosenkrantz, D. J., & Tayi, G. K. (2018). Approximation algorithms for facility dispersion.**
>   As shown in the title, this paper explicitly emphasized facility location as a strong motivation, and studied multiple dispersion objectives, including both *max-min diversification* and *max-sum diversification*.
> - **Ploskas, N., Stergiou, K., & Tsouros, D. C. (2023). The p-dispersion problem with distance constraints.**
>   This recent work further emphasized the application of *p*-dispersion (*max-min diversification*) in facility location, and investigated the *p*-dispersion with distance constraints (PDDP), a natural variant where hard minimum distance requirements between facilities are imposed.
>
> These and many related papers consistently highlight that diversity maximization provides a natural framework for facility location, especially in commercial or service applications where spreading out facilities helps maximize customer flow and economic benefits.
>
> At the same time, recent research on **individual fairness** in facility location (e.g., Vakilian and Yalçıner (2022), *Improved Approximation Algorithms for Individually Fair Clustering*; Jung et al. (2019), *Service in Your Neighborhood: Fairness in Center Location*) highlights the complementary motivation of ensuring that each individual in the dataset lies within a reasonable distance of some facility. Thus, combining diversity maximization with individual fairness yields a powerful formulation: from the perspective of decision-makers, facilities are dispersed to avoid harmful concentration and maximize reach, while from the perspective of individuals, no one is left underserved.
>
> We believe this dual motivation is both standard and compelling, and we will revise the manuscript to add the above references so that readers can clearly see this connection.

---

> > ### Comment · Reviewer_uok7 · 2025-08-04
> >
> > I thank the authors for the response. My recommendation is that the draft should include these motivations based on facility location and acknowledge the possible exponential dependence on $k$. If these two things are met, I will lift my score to 4.

---

> > > ### Author Response · Authors · 2025-08-05
> > > **Thank You for the Constructive Suggestions and Support!**
> > >
> > > We sincerely thank you for your constructive feedback! Your suggestions have indeed helped us make our work more complete. We will incorporate the motivations based on facility location and acknowledge the possible exponential dependence on $k$ in the revised version of the paper.
> > >
> > > Thank you again for your thoughtful review and support!

---

### Official Review · Reviewer_NtvU · 2025-07-03

**Clarity:** 3
**Significance:** 3
**Originality:** 2
**Rating:** 4
**Confidence:** 3

**Summary:**

In this paper, the authors study the problem of diversity maximization under individual fairness constraints. The objective is to select a subset of $k$ points from a dataset such that diversity is maximized while ensuring fairness, where every point has a representative within a bounded distance. The authors provide a $(O(1), 3)$ bicriteria approximation algorithm, where the representative is at most $3$ times the distance to the nearest neighbor, and diversity is a constant fraction of the optimal diversity. These algorithms leverage fairness-aware partitioning techniques and matroid-based optimization to achieve theoretical guarantees for fairness and diversity. Empirical evaluations on real-world and synthetic datasets validate their approach.

**Questions:**

Can the authors comment on whether diversity maximization with individual fairness constraints versus group fairness constraints is equivalent in terms of computational hardness? Or is there any reduction between these problems?

**Ethical Concerns:**

["NO or VERY MINOR ethics concerns only"]

**Final Justification:**

I'm happy with rebuttal response. As I asked for clarification questions, I am keeping my score (borderline accept).

**Quality:**

3

**Strengths And Weaknesses:**

**Strengths**:

While group fairness has been well studied, individual fairness with diversity-based objectives has not been studied before. The motivation of the setting is novel and an important addition to the diversity maximization literature. The authors provide approximation algorithmic results for different diversity maximization objectives studied in the literature with individual fairness.

Many prior works (on group fairness) design algorithms by reduction to partition matroids. In this work, the authors design an interesting approach of partitioning the points into individual fairness regions and then employing partition matroid methods to satisfy the constraints.

**Weaknesses**:

The algorithmic novelty is limited. It would be interesting to see if there is a way to improve on the $O(1)$ guarantees. It suggests that the guarantee is because of the partition matroid, and it is not immediately clear how hard the problem is with respect to group fairness.

The computational complexity of $O(n^2)$ is certainly a challenge when scaling up to real-world datasets. It would be interesting to understand if there is a hardness result (fine-grained complexity) with regard to computational complexity.

---

> ### Author Rebuttal · Authors · 2025-07-30
>
> We sincerely thank you for your valuable feedback and the time you dedicated to reviewing our paper! We address each of your comments point by point below.
>
> > Q1: The algorithmic novelty is limited.
>
> A1: To the best of our knowledge, our work is the first in the literature to integrate individual fairness guarantees with diversity objectives. Moreover, rather than focusing on a single diversity objective, we systematically analyze three well-established diversity functions, max-min, max-sum, and sum-min, which have each received significant attention in prior work. This broader scope requires distinct algorithmic designs, analyses and proofs tailored to the structural differences of these objectives. We therefore believe that both the problem formulation and the algorithmic contributions reflect meaningful novelty.
>
>
>
> > Q2: It would be interesting to see if there is a way to improve on the $O(1)$ guarantees.
>
> A2: We appreciate the reviewer's comment on potential improvements over the current $O(1)$ approximation guarantees. However, asymptotic improvements are not possible. We now prove that for all three diversity maximization objectives under individual fairness constraints, no polynomial-time algorithm can achieve a diversity approximation factor better than $O(1)$. The proof is similar to that in Moumoulidou et al. (2020) *Diverse Data Selection under Fairness Constraints*.
>
> **Theorem 1** There exists no polynomial-time $\beta$-approximation algorithm for individually fair max-min diversification with $\beta<2$, unless P=NP.
>
> *Proof Sketch.* Suppose that there exists a polynomial algorithm that approximates the diversity score of the optimal solution to individually fair max-min diversification by a factor of $\beta<2$. Then, this algorithm could also solve the unconstrained max-min diversification problem with approximation factor $\beta$. However, Ravi et al. (1994) *Heuristic and special case algorithms for dispersion problems* have shown that unconstrained max-min diversification cannot be approximated within a factor better than 2, through a reduction from the clique problem. Therefore, it is not possible for such an algorithm to exist.
>
> By a similar argument, and leveraging the fact that max-sum and sum-min diversification cannot be approximated within a factor better than 2 under the planted clique conjecture (Bhaskara et al. (2016) *Linear relaxations for finding diverse elements in metric spaces*), we can also prove the following theorem:
>
> **Theorem 2** There exists no polynomial-time $\beta$-approximation algorithm for individually fair max-sum and sum-min diversification with $\beta<2$ under planted clique conjecture.
>
> Therefore, the unconstrained diversity maximization problems can be viewed as special cases of our setting under individual fairness constraints. This implies that our problems cannot achieve asymptotic improvements beyond the known $O(1)$ approximation barrier. Nevertheless, future work may still refine these results by establishing smaller constant-factor bounds. We will add the above two theorems and their proofs to the paper to explicitly demonstrate the hardness of our problem.
>
>
>
> > Q3: It suggests that the guarantee is because of the partition matroid, and it is not immediately clear how hard the problem is with respect to group fairness.
>
> A3: In our algorithm, the individual fairness constraint is transformed into a partition matroid constraint, and therefore, the approximation guarantees under individual fairness are directly related to those under partition matroid constraints. By definition, any solution that satisfies group fairness also satisfies the partition matroid constraint, making partition matroids a more general class, with group fairness being a specific instantiation. This implies that solving the problem under group fairness is a more specific and arguably simpler case.
>
> However, it is important to note that individual fairness and group fairness are not directly interchangeable: they capture fundamentally different notions of representation, and results for one do not automatically extend to the other.
>
>
>
> > Q4: The computational complexity of $O(n^2)$ is certainly a challenge when scaling up to real-world datasets. It would be interesting to understand if there is a hardness result (fine-grained complexity) with regard to computational complexity.
>
> A4: We agree that, in the current implementation of Algorithm 1 and Algorithm 2, the computational bottleneck arises from an $O(n^2 \log n)$ time complexity. This cost is primarily incurred due to repeatedly identifying the minimum fair radius among all remaining elements (Line 4 in Algorithm 1). In our prototype implementation, we employed Python’s built-in `set` data structure and performed sorting during each iteration to identify the minimal element, which indeed leads to the stated complexity.
>
> However, as the reviewer correctly noted, this is not an inherent lower bound of the algorithm but rather a byproduct of our current implementation strategy. In practical systems with efficiency concerns (such as in industrial applications), more sophisticated data structures could be employed to significantly reduce the computational overhead. For example:
>
> - If we store the candidate elements in a list or array, we can sort them once at the beginning, and the subsequent set difference operations will not affect the internal order. This allows for an overall $O(n \log ⁡n)$ time complexity.
> - Alternatively, using a min-heap or a priority queue enables extracting the minimum fair radius in $O(\log⁡n)$ time per iteration. Since the outer loop runs at most $n$ times, this results in an improved total complexity of $O(n\log ⁡n)$.
>
> Given that the time complexity can theoretically be reduced, in the revised version we will optimize the implementation accordingly and add the above discussion to clarify this point (Section: **B.1 Proof of Lemma 1**).
>
>
>
> > Q5: Can the authors comment on whether diversity maximization with individual fairness constraints versus group fairness constraints is equivalent in terms of computational hardness?
>
> A5: At present, analyses of the computational hardness of diversity maximization under either group fairness or individual fairness constraints are derived from results in the unconstrained setting. Specifically, if $P \neq NP$, then no polynomial-time algorithm can achieve an approximation factor better than 2 for max-min diversification; and under the planted clique assumption, the same hardness holds for max-sum and sum-min diversification. These results imply that, regardless of whether group fairness or individual fairness constraints are imposed, no polynomial-time algorithm is expected to surpass the factor-2 barrier. You may refer to our response to Q2, where we formally present and prove that for all three diversification objectives under individual fairness constraints, the approximation factor has a lower bound of 2.
>
> Moreover, we emphasize that after defining the fairness parameter $\alpha$ in Definition 2, we explicitly note in Lines 152–155 that even finding a feasible solution that strictly satisfies the individual fairness constraint is itself NP-hard. This further underscores the additional difficulty of diversity maximization under individual fairness constraints.
>
>
>
> > Q6: Is there any reduction between these problems?
>
> A6: Yes, there is. The unconstrained diversity maximization problem can be seen as a special case of individually fair diversity maximization, where the fairness parameter $\alpha$ is sufficiently large so that the fairness constraints become vacuous. A more detailed argument along this line is provided in our response to Q2.
>
>
>
> Thank you again for your constructive comments!

---

> > ### Comment · Reviewer_NtvU · 2025-08-05
> > **Reviewer Response**
> >
> > I thank the authors for answering my questions and request the authors to update the version with the response provided.

---

> > > ### Author Response · Authors · 2025-08-05
> > > **Thank You for the Constructive Suggestions and Support!**
> > >
> > > We sincerely thank you for your thoughtful comments, which have made our work more complete! At the current rebuttal and discussion stage, we are not allowed to immediately update the paper. However, if revisions are permitted, we will incorporate the updates in the revised version.
> > >
> > > Specifically:
> > >
> > > - We will add a new **Hardness of Approximation** section between *Problem Definition* and *Our Algorithms*, where we will include the theorem and proof from A2 of our rebuttal to explicitly state a lower bound on the approximation factor and clarify its reduction relationship to the unconstrained case.
> > > - We will revise **Lemma 1** and **Appendix B.1 (Proof of Lemma 1)** to correct the time complexity of Algorithm 1, and update the related statement in Algorithm 2, improving the bound from $O(n^2 \log n)$ to $O(n \log n)$.
> > > - If space is limited, we will provide the detailed discussion and proofs in the **Appendix**, to ensure that all technical details are fully documented.
> > >
> > > Thank you again for your valuable suggestions, which help us significantly improve our work!

---

### Note · Authors · 2025-08-11

We sincerely thank the reviewers for their detailed and constructive feedback, and also express our gratitude to the Area Chair and Senior Area Chair for their time and oversight. Here, we would like to summarize the main points addressed during the rebuttal process.

We appreciate the reviewers’ recognition of several aspects of our work. After further clarifying the motivation, all reviewers acknowledged the relevance and practicality of combining individual fairness with diversity maximization. Some also appreciated our algorithms, writing clarity, and overall solidity.

Reviewers also pointed out areas for potential improvement, to which we responded in detail. Regarding experiments, we emphasized the presence of suitable baselines and added experiments that further motivate the problem formulation, highlighting that our method can effectively address the given problem. On the theoretical side, some reviewers observed potential issues with the time complexity. We refined the complexity for Algorithms 1 and 2 from $O(n^2 \log n)$ to $O(n \log n)$, improving practical applicability. For Theorem 4, we discussed the trade-offs among metric space structure, strict satisfaction of constraints, approximation factor, and time complexity, explaining why the latter cannot currently be theoretically guaranteed without affecting the others, while our experiments (Figure 3) show practical performance.

Some reviewers were curious about our frequent connections between individual fairness, partition matroids, and group fairness. We clarified that any solution satisfying group fairness constraints also satisfies partition matroid constraints, but individual fairness and group fairness are inherently non-interchangeable, motivating our use of partition matroids for problem reduction.

We acknowledge that, in terms of hardness of approximation, our current work does not establish tighter bounds for the three problems discussed in the paper. However, we show that no polynomial-time algorithm can achieve an approximation factor better than 2 in our setting. This implies that future work aiming for stronger approximation guarantees cannot improve the asymptotic bound, but may focus on improving the constant factor.

The reviewers’ comments have been invaluable for improving our work. If possible, we will incorporate all suggested changes in a revised version. We again thank everyone involved in the review and discussion process for their time, effort, and insights.

---

### Decision · Program_Chairs · 2025-09-17

**Decision:**

Accept (poster)

**Comment:**

This paper addresses diversity maximization under individual fairness constraints, which is a novel and timely extension of prior fairness work. Reviewers found the formulation clear and the approximation guarantees interesting, and they appreciated the contributions as relevant to the fairness/diversity literature. However, concerns were raised regarding:

•	Motivation and applications, which need to be made clearer and illustrated with stronger examples.

•	Computational complexity, especially the exponential dependence in Theorem 4, which should be explicitly discussed.

•	Positioning against prior work and comparisons to baselines, which are currently limited.

Overall, the paper makes a meaningful contribution, though it would benefit from clearer motivation, computational complexity discussion in the final version. We strongly encourage authors to address reviewers' feedback in the final version of their paper.